

# Boundary modes in the Chamon model

**Weslei B. Fontana**[1*] **and Rodrigo G. Pereira**[1,2]

**1** International Institute of Physics, Universidade Federal
do Rio Grande do Norte, 59078-400 Natal, RN, Brazil
**2** Departamento de Física Teórica e Experimental, Universidade Federal
do Rio Grande do Norte, 59072-970 Natal, RN, Brazil

⋆ weslei.fontana@ufrn.br

## Abstract

We study the fracton phase described by the Chamon model in a manifold with a boundary. The new processess and excitations emerging at the boundary can be understood by means of a diagrammatic framework. From a continuum perspective, the boundary theory is described by a set of scalar fields in similarity with standard $K$-matrix Chern-Simons theory. The continuum theory recovers the gapped boundaries of the lattice model once we include sufficiently strong interactions that break charge conservation. The analysis of the perturbative relevance of the leading interactions reveals a regime in which the Chamon model can have a stable gapless fractonic phase at its boundary.

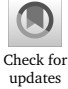
# 1 Introduction

Fracton phases [1–25] constitute an intriguing and novel type of quantum matter, whose understanding sits at a confluence of different research fields. Fractonic phases are often classified in terms of the restricted mobility presented by their emergent quasiparticles, which can either move under mild restrictions (type-I) or are completely immobile (type-II) due to the proliferation of new excitations at every step of their motion. Three-dimensional gapped fracton phases also display a robust ground state degeneracy. However, unlike conventional two-dimensional (2D) topological phases, their ground state degeneracy depends on some geometric data, typically growing exponentially with the linear size of the system.

Fracton systems have been studied in several distinct setups. They can be obtained through the extension of 2D topological order to three dimensions, either by directly constructing 3D spin models [1–5] or via stacking of 2D topological phases [10–15, 26–30]. Fractonic phases can also be described by effective continuum theories, usually involving higher-rank gauge fields or exotic Chern-Simons theories [9, 17–21, 31–35]. In the spirit of the latter, it is natural to ask whether a fractonic system can exhibit nontrivial boundary phenomena in analogy with 2D topological phases, where different gapped boundaries can be classified by the condensation of quasiparticles and are separated by quantum phase transitions [36–42].

Boundary theories of fracton phases may present peculiar properties due to the geometric sensitivity natural to these systems. In the context of lattice theories, gapped boundaries of the $\mathbb{Z}_2$ X-cube model [3, 8] have been analyzed in Refs. [43, 44]. It has been pointed out that braiding of fractons in the bulk is geometry dependent and insufficient to classify the different types of boundaries [43]. More recently, the authors of Ref. [45] approached the X-cube model from a continuum perspective and showed that its boundary theory can be viewed as a generalized $K$-matrix Chern-Simons theory resembling the situation in standard topological orders [46–48].

In this work, we study the boundary theory for the $\mathbb{Z}_2$ Chamon model [1] from both perspectives, lattice and continuum. First we show that the excitations in the lattice model can be represented by a combination of a few fundamental diagrammatic structures, closely related to cage nets in other fracton models [49, 50]. At the boundary, additional diagrams are allowed and account for new processes that violate fracton parity constraints and affect the mobility of the excitations.

On the continuum side, we extend on previous work [20] and show that the Chamon model can be described by a Chern-Simons-like theory with higher derivatives and two gauge fields labeled by a layer index. In the presence of a boundary, the effective field theory gives rise to physical boundary degrees of freedom described by a $K$-matrix theory. The boundary action is equivalent to the one recently derived for the X-cube model [45], despite the different bulk excitation spectrum of these models. However, unlike the X-cube model, the description of a (001) boundary in the Chamon model requires imposing a boundary condition for the normal derivative of the fields. As one of our main results, we show that our prescription leads to a consistent boundary theory with the correct number of degrees of freedom and encodes the information that line operators that terminate at the boundary create a single fracton at the endpoint.

To recover the gapped boundary spectrum of the lattice model, we investigate the effects of perturbations allowed by the discrete symmetry and the compactification of the bosonic fields. While the boundary phases of the lattice model can be identified with the strong coupling regime, we also discuss the perturbative relevance of the interactions in the weak coupling regime. Another main result is that we predict the existence of a stable gapless boundary phase where all perturbations that break charge conservation become irrelevant. Such a phase has emergent continuous subsystem symmetries and is unexpected from the point of view of the microscopic model.

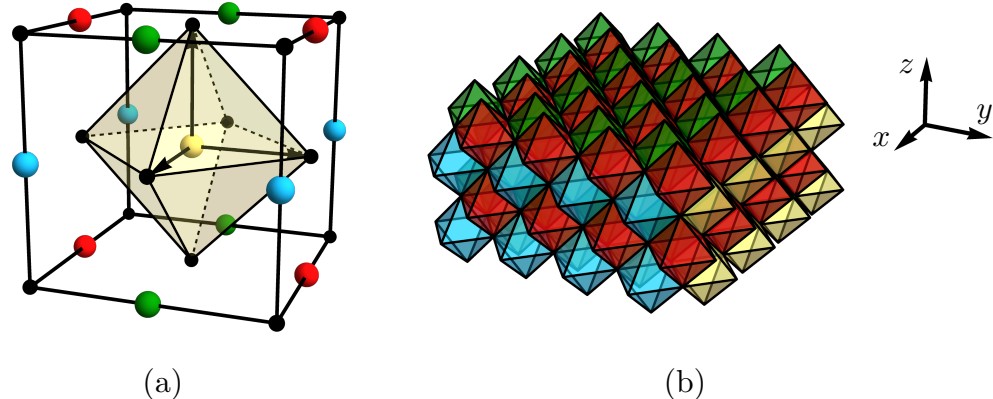

(a)                                                        (b)

Figure 1: Chamon model on the fcc lattice. (a) Black dots represent sites in $\Omega_{\text{even}}$ where the spin operators act. The positions in $\Omega_{\text{odd}}$ are divided into A, B, C and D sublattices, represented by the colors red, green, blue, and yellow, respectively. Every position in $\Omega_{\text{odd}}$ is associated with an octahedron and the corresponding stabilizer defined as in Eq. (1). (b) fcc lattice of octahedra. We distinguish between two types of (001) planes, in which the octahedra belong to either A and B or to C and D sublattices.

The paper is organized as follows. In Sec. 2 we present the $\mathbb{Z}_2$ Chamon model both in the bulk and in the presence of a boundary. In Sec. 3 we review the formalism of Ref. [20] and examine the symmetries of the effective field theory. Section 4 contains our results on the continuum description of the boundary modes. In Sec. 5 we offer some final remarks and point out possible routes for future work. Finally, the diagonalization of the boundary Hamiltonian and the calculation of correlation functions of charged operators are detailed in the appendices.

## 2 Lattice model

The Chamon code [1, 2] is an exactly solvable spin model defined on the fcc lattice which displays type-I fractonic behavior. Consider a cubic lattice $\Omega$ with sublattices $\Omega_{\text{even}}$ and $\Omega_{\text{odd}}$. We say that a given site $\mathbf{r} = a_s(i, j, k)$, with $i, j, k \in \mathbb{Z}$ and $a_s$ the lattice spacing, belongs to $\Omega_{\text{even}}$ ($\Omega_{\text{odd}}$) if $i + j + k$ is even (odd). We assume that both $\Omega_{\text{even}}$ and $\Omega_{\text{odd}}$ contain an even number of sites. We place a spin 1/2 with Pauli operators $\sigma_{\mathbf{r}}^I$, where $I \in \{x, y, z\} \equiv \{1, 2, 3\}$, at every site $\mathbf{r} \in \Omega_{\text{even}}$. For every $\mathbf{r} \in \Omega_{\text{odd}}$ we define the stabilizer operator acting on the vertices of the octahedron centered at $\mathbf{r}$ [see Fig. 1(a)]:

$$\mathcal{O}_{\mathbf{r}} = \sigma_{\mathbf{r}+\hat{\mathbf{e}}_1}^x \sigma_{\mathbf{r}-\hat{\mathbf{e}}_1}^x \sigma_{\mathbf{r}+\hat{\mathbf{e}}_2}^y \sigma_{\mathbf{r}-\hat{\mathbf{e}}_2}^y \sigma_{\mathbf{r}+\hat{\mathbf{e}}_3}^z \sigma_{\mathbf{r}-\hat{\mathbf{e}}_3}^z, \tag{1}$$

where $\hat{\mathbf{e}}_1 = a_s(1, 0, 0)$, $\hat{\mathbf{e}}_2 = a_s(0, 1, 0)$, and $\hat{\mathbf{e}}_3 = a_s(0, 0, 1)$. The Hamiltonian of the Chamon model is given by

$$H = -\sum_{\mathbf{r} \in \Omega_{\text{odd}}} \mathcal{O}_{\mathbf{r}}. \tag{2}$$

Since all stabilizers commute among themselves and square to the identity, $\mathcal{O}_{\mathbf{r}}^2 = \mathbf{1}$, the model is exactly solvable with eigenstates labeled by the eigenvalues $\pm 1$ of the stabilizers. A ground state $|\Psi\rangle$ of the Hamiltonian obeys the condition

$$\mathcal{O}_{\mathbf{r}} |\Psi\rangle = |\Psi\rangle, \quad \forall \, \mathbf{r} \in \Omega_{\text{odd}}. \tag{3}$$

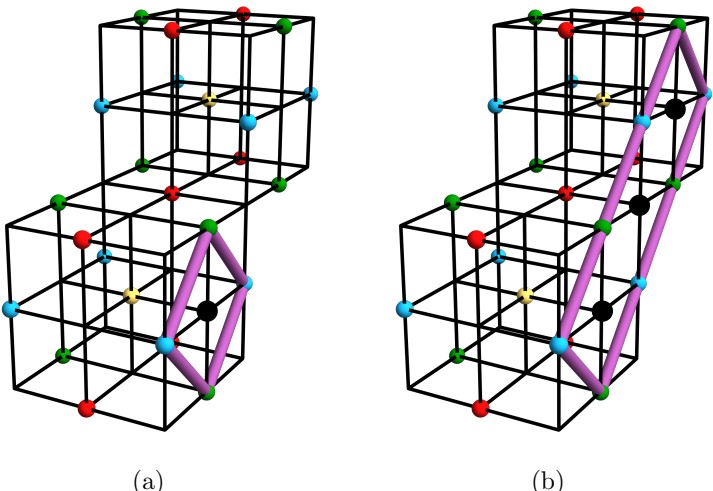

Figure 2: Diagrammatic representation of excitations. (a) The action of $\sigma_{\mathbf{r}}^x$ on the site marked by a black dot flips the sign of four stabilizers in a $yz$ plane with B and C sublattices. The quadrupole is represented by strings connecting the defects. (b) Successive application of spin operators in the same plane can stretch the strings and move the dipoles in the direction of either $\hat{\mathbf{e}}_2 + \hat{\mathbf{e}}_3$ or $\hat{\mathbf{e}}_2 - \hat{\mathbf{e}}_3$. The excitations reside at the corners of the ribbon structure.

The ground state degeneracy of the Chamon model on a 3-torus was calculated in Ref. [2]. The ground states can be labeled by the eigenvalues of rigid line operators that wind around the lattice. If all lattice dimensions are even, with lengths $L_x = 2p_x$, $L_y = 2p_y$, and $L_z = 2p_z$, the number of ground states is $2^{4\gcd(p_x,p_y,p_z)}$ where $\gcd(p,q,r)$ stands for the greatest common divisor of the integers $p,q,r$.

We refer to a single defect stabilizer with $\mathcal{O}_{\mathbf{r}} = -1$ as a fracton. In a lattice with periodic boundary conditions in all three directions, the model has the constraints

$$\prod_{\mathbf{r}\in A} \mathcal{O}_{\mathbf{r}} = \prod_{\mathbf{r}\in B} \mathcal{O}_{\mathbf{r}} = \prod_{\mathbf{r}\in C} \mathcal{O}_{\mathbf{r}} = \prod_{\mathbf{r}\in D} \mathcal{O}_{\mathbf{r}} = \mathbf{1}, \tag{4}$$

where A, B, C, D denote the four sublattices of $\Omega_{\text{odd}}$ represented in Fig. 1(a) as red, green, blue, and yellow dots, respectively. More specifically, we label the sublattices as follows:

$$\begin{aligned} &\text{A:} \quad \mathbf{r} = a_s(2l+1, 2m, 2n), &\quad &\text{B:} \, \mathbf{r} = a_s(2l, 2m+1, 2n), \\ &\text{C:} \quad \mathbf{r} = a_s(2l, 2m, 2n+1), &\quad &\text{D:} \, \mathbf{r} = a_s(2l+1, 2m+1, 2n+1), \end{aligned} \tag{5}$$

where $l, m, n \in \mathbb{Z}$. Equation (4) implies that defects can only appear in pairs on each of the sublattices. In fact, the action of a Pauli operator on a ground state creates four defects that belong to two sublattices. For instance, the operator $\sigma_{\mathbf{r}}^z$ anticommutes with four stabilizers contained in a (001) plane, namely $\mathcal{O}_{\mathbf{r}\pm\hat{\mathbf{e}}_1}$ and $\mathcal{O}_{\mathbf{r}\pm\hat{\mathbf{e}}_2}$. Applying $\sigma_{\mathbf{r}}^z$ on the ground state, we create four defects belonging to A and B sublattices if $\mathbf{r}\cdot\hat{\mathbf{e}}_3$ is even or to C and D sublattices if $\mathbf{r}\cdot\hat{\mathbf{e}}_3$ is odd; see Fig. 1(b). Importantly, the independent $\mathbb{Z}_2$ constraints in Eq. (4) allow us to distinguish between two sets of (001) planes (and equivalently for the other spatial directions) and to label the defects by the pair of sublattices on which they are created.

The model admits a diagrammatic description similar to that of cage-net fracton models [49]. We start by representing the action of a local spin operator by strings connecting the four

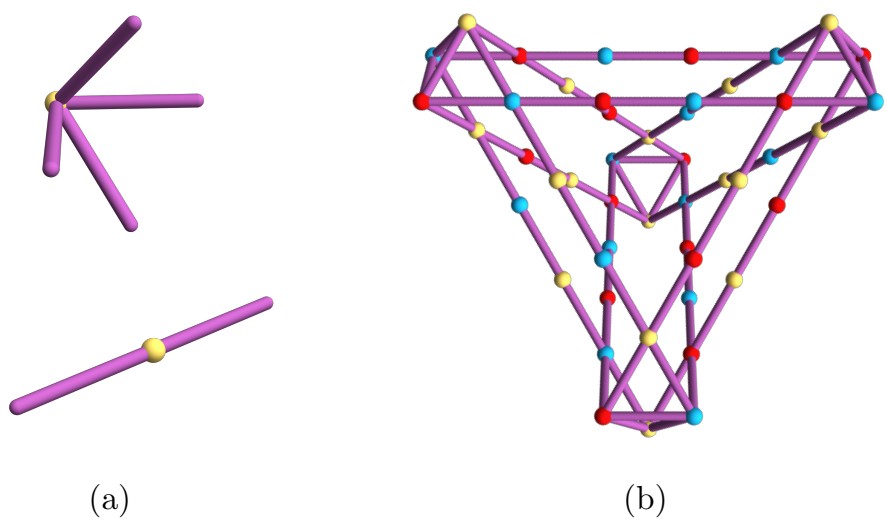

(a)                                        (b)

Figure 3: (a) Basic diagrams representing the string configurations allowed in the ground state sector. (b) Example of a cage-net tetrahedral structure.

flipped stabilizers on a given plane, see Fig. 2(a). Due to the $\mathbb{Z}_2$ character of the model, the superposition of two strings on a link between two stabilizers is equivalent to no string. Thus, applying spin operators along a line we can create an enlarged ribbon structure that accounts for the motion of pairs of defects as shown in Fig. 2(b). In this representation, the defects can be identified as the corners at which the strings form 90° angles. Following the nomenclature of Ref. [2], we will refer to the pairs of defects at the ends of the strings as dipoles. Note that in this $\mathbb{Z}_2$ model the dipole is made up of two defects with the same $\mathbb{Z}_2$ charge, but these defects cannot annihilate each other due to the constraints in Eq. (4). Importantly, these dipoles move along rigid lines aligned with face diagonals of the lattice. By contrast, a fracton (or monopole) cannot move without creating additional excitations, whereas there are no mobility constraints for four defects (a quadrupole) which can be created by the action of local operators on the ground state. This nomeclature is further motivated by the higher-moment conservation laws in U(1)-symmetric effective field theories for fracton models [51, 52]. We will discuss the gauge theory for the Chamon model in Sec. 3.

We can now ask about all the basic diagrams representing operators which commute with the Hamiltonian. By construction, the action of such operators on a ground state must preserve the condition in Eq. (3). The allowed configurations are shown in Fig. 3(a). The first type of diagram consists of a vertex with four strings emanating from the same site and forming 90° corners in two different planes. The second type is simply a straight line passing through a stabilizer site. From these basic elements we can build cage nets without any defects at their endpoints, as shown in Fig. 3(b). This type of cage-net tetrahedron in the lattice model is consistent with the form of gauge-invariant operators in the Chern-Simons theory as discussed in Ref. [9].

We now turn to the model with a boundary. We consider a semi-infinite volume with an open boundary placed at $z = 0$, with the system occupying the region $z \leq 0$. At the boundary

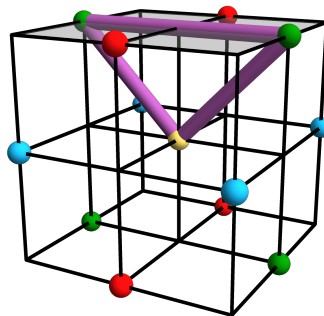 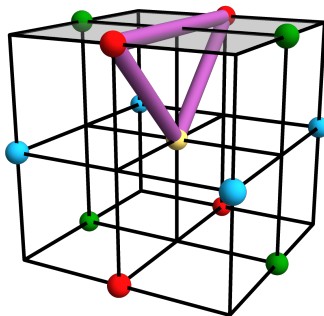

Figure 4: A (001) boundary that breaks the parities associated with C and D sublattices. The broken symmetries allow for triangular diagrams contained in $xz$ and $yz$ planes.

we define "broken" stabilizers given by five-site operators

$$\mathcal{O}_{\mathbf{r}} = \sigma^{x}_{\mathbf{r}+\hat{\mathbf{e}}_1} \sigma^{x}_{\mathbf{r}-\hat{\mathbf{e}}_1} \sigma^{y}_{\mathbf{r}+\hat{\mathbf{e}}_2} \sigma^{y}_{\mathbf{r}-\hat{\mathbf{e}}_2} \sigma^{z}_{\mathbf{r}-\hat{\mathbf{e}}_3}, \tag{6}$$

for $\mathbf{r} = a_s(i, j, 0) \in \Omega_{\text{odd}}$. We refer to a single defect of a five-site stabilizer as a boundary fracton. Physically, we can understand this choice of the boundary stabilizers by considering that we start from the Chamon model in the thermodynamic limit and apply a strong magnetic field in the $z$ direction, $H_Z = -h \sum_{\mathbf{r}, z>0} \sigma^z_{\mathbf{r}}$, to freeze out the spins in the half space $z > 0$. The effective Hamiltonian is then obtained by projecting the stabilizers in the $z \leq 0$ region onto the sector where all spins in the $z > 0$ region are polarized with $\sigma^z_{\mathbf{r}} = +1$. On the boundary plane, this projection reduces the standard octahedral stabilizers in Eq. (1) to the five-site stabilizers in Eq. (6).

The boundary stabilizers violate two out of the four constraints in Eq. (4). For the boundary at $z = 0$, the parities associated with C and D sublattices are no longer conserved. Similarly, shifting the boundary to $z = -1$ would break the parities associated with A and B sublattices. Note that, unlike the X-cube model [43, 44], removing a layer in the Chamon model does not alternate between "smooth" and "rough" boundaries in analogy with the terminology for the toric code [38]. In contrast, the boundaries at $z$ even and $z$ odd are geometrically similar, but they can be classified by the pair of constraints that they break.

From the action of spin operators on stabilizers close to the boundary, we obtain new string configurations in the diagrammatic description. The new basic diagrams take the form of triangles in $xz$ and $yz$ planes with the longest edge at the boundary, as depicted in Fig. 4. Importantly, the creation of an odd number of defects out of the vacuum is now possible due to the broken parities for two sublattices.

The triangular diagrams are associated with distinctive processes involving the excitations that reach the boundary. The simplest example is the transmutation of a dipole into a single defect. We first bring the dipole to the boundary by successive application of spin operators along a line, as depicted in Fig. 5(a). We then glue the appropriate triangle, which has the effect of converting the pair of defects into a fracton, as in Fig. 5(b). The single defect is now located at the corner of the ribbon that forms a 45° angle. Note that the remaining boundary fracton belongs to a sublattice for which the constraint in Eq. (4) is preserved by the boundary conditions. If we label the defects by $\nu = a, b, c, d$ corresponding to A, B, C, D sublattices, respectively, the bulk excitations fuse into the boundary at $z = 0$ as $(a, c) \rightarrow a$, $(a, d) \rightarrow a$, $(b, c) \rightarrow b$, and $(b, d) \rightarrow b$. On the other hand, the boundary at $z = -1$ allows for the fusion $(a, c) \rightarrow c$, $(a, d) \rightarrow d$, $(b, c) \rightarrow c$, and $(b, d) \rightarrow d$.

Another important process is the reflection of a dipole at the boundary. After converting the dipole into a boundary fracton, we can again compose the diagram with another triangle that

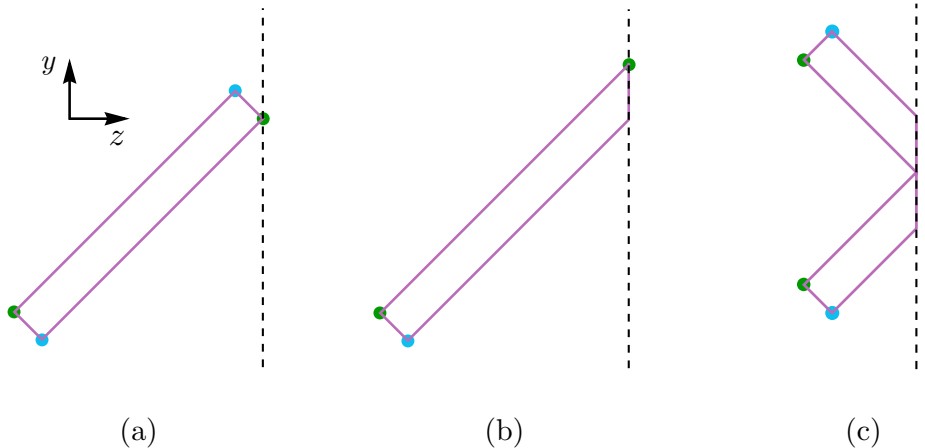

(a)                          (b)                          (c)

Figure 5: Processes involving dipoles that propagate to the boundary. (a) A dipole can be brought from the bulk to the boundary through successive combinations of the square diagrams in Fig. 2. (b) The dipole can be converted into a boundary fracton using the triangular diagrams in Fig. 4. (c) The dipole can be reflected at the boundary and change its direction of propagation. This process also implies that a pair of dipoles moving in different directions can annihilate on the boundary if the endpoints of the ribbons coincide.

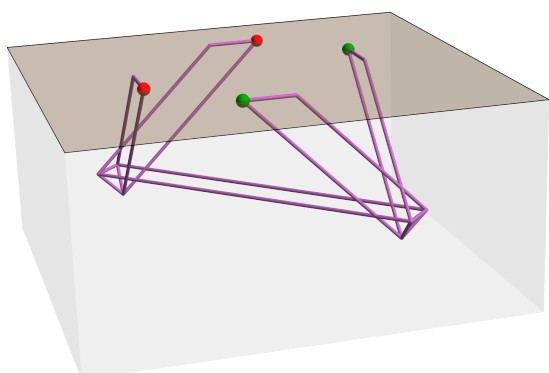

Figure 6: A boundary half cage for the (001) boundary at $z = 0$. The red and green dots mark the positions of the isolated boundary fractons.

transforms the fracton back into a dipole, but now propagating in a perpendicular direction in the same or in a different plane. The diagram for reflection in the same plane is shown in Fig. 5(c). Alternatively, we can view the diagram in Fig. 5(c) as a process in which two dipoles meet at the boundary and annihilate. Importantly, there is no process which allows for a single dipole to condense at a (001) boundary of the Chamon model. In contrast, pairs of $e$ or $m$ particles can condense in the rough or smooth boundaries of the X-cube model [43,44].

The possibility of converting a bulk dipole into a boundary fracton also modifies the mobility of dipoles along the surface. When an $(a, b)$ dipole in the $z = 0$ plane meets a bulk dipole that impinges on the boundary, the bulk dipole can be reflected back while the dipole that moves on the boundary plane changes its direction of propagation. Thus, a boundary dipole can lift its mobility restrictions by undergoing an elastic collision with a bulk dipole.

Finally, we can associate boundary half cages (BHCs) [43] with the (001) boundaries of the Chamon model. By definition, a BHC operator does not create any excitations in the bulk but creates isolated excitations on the boundary. We construct such an operator by bringing

a cage-net tetrahedron to the boundary and terminating four ribbons with boundary fractons. Once again, the pair of constraints respected by the boundary determines the sublattice indices for the isolated monopoles that can appear in the BHCs. A particular example is illustrated in Fig. 6.

# 3 Effective field theory in the bulk

The physics of the Chamon model can be understood from a effective theory perspective by means of a Chern-Simons-like theory. A first attempt of describing the model in this way through a top-down approach can be found in [9] and later on a description of the model via a bottom-up approach and the generalization of it in higher dimensions was obtained in [20]. In this section we review the framework of Ref. [20] and argue about the necessity of a layer index, that is explicit in the microscopic description, that has an important effect in the continuum theory of the bulk, leading to a BF-like theory instead of the previously Chern-Simons-like descriptions.

## 3.1 Effective action

We start by representing the Pauli operators via the exponential map

$$\sigma_{\mathbf{r}}^I \sim \exp\left[i\, t_m^I K_{mn} \theta_n(\mathbf{r})\right],\tag{7}$$

where we assume an implicit sum over repeated indices $m, n \in \{1, 2\}$. Here $K$ is the $2 \times 2$ antisymmetric matrix

$$K = \begin{pmatrix} 0 & 1 \\ -1 & 0 \end{pmatrix}.\tag{8}$$

The Pauli operators are expressed in terms of two lattice fields $\theta_n(\mathbf{r})$. The identity $\sigma_{\mathbf{r}}^x \sigma_{\mathbf{r}}^y \sigma_{\mathbf{r}}^z = i\mathbf{1}$ implies a neutrality condition for the $t$-vectors, $\sum_{I=1}^3 t_m^I = 0$. We can single out the $z$ direction and fix the $t$-vectors simply as

$$t_m^I = \begin{cases} \delta_m^I, & I = 1, 2, \\ -1, & I = 3. \end{cases}\tag{9}$$

The algebra of the Pauli operators translates into the new variables through the relations

$$t_m^I K_{mn} t_n^J = \begin{cases} 0 \pmod{2}, & \text{if } I = J, \\ 1 \pmod{2}, & \text{if } I \neq J, \end{cases}\tag{10}$$

provided that we impose the commutation relations

$$\left[\theta_m(\mathbf{r}), \theta_n(\mathbf{r}')\right] = i\pi \left(K^{-1}\right)_{mn} \delta_{\mathbf{r},\mathbf{r}'}.\tag{11}$$

The lattice fields are compact since the shift $\theta_n(\mathbf{r}) \to \theta_n(\mathbf{r}) + 2\pi m_n$ with $m_n \in \mathbb{Z}$ leaves the Pauli operators invariant.

The representation with the $t$-vectors in Eq. (9) is spatially anisotropic. Given the special role of the $z$ direction, we label the fields at each point by whether $\mathbf{r} \in \Omega_{\text{odd}}$ belongs to the set $\mathcal{L}_1$ of (001) planes with A and B sublattices ($\mathbf{r} \cdot \hat{\mathbf{e}}_3$ even) or to the set $\mathcal{L}_2$ of planes with C and D sublattices ($\mathbf{r} \cdot \hat{\mathbf{e}}_3$ odd). We introduce a layer index $l$ and write $\theta_n(\mathbf{r}) \to \theta_n^{(l)}(\mathbf{r})$, with $l = 1$ for AB planes and $l = 2$ for CD planes. We also define the forward and backward lattice derivatives

$$\overrightarrow{\Delta}_I \theta_n^{(l)}(\mathbf{r}) = \theta_n^{(l)}(\mathbf{r} + \hat{\mathbf{e}}_I) - \theta_n^{(l)}(\mathbf{r}),\tag{12}$$

$$\overleftarrow{\Delta}_I \theta_n^{(l)}(\mathbf{r}) = \theta_n^{(l)}(\mathbf{r}) - \theta_n^{(l)}(\mathbf{r} - \hat{\mathbf{e}}_I),\tag{13}$$

along with the second derivative $\Delta_I^2 = \overrightarrow{\Delta}_I \overleftarrow{\Delta}_I$. The lattice Hamiltonian in Eq. (2) can then be written as

$$H = -\sum_{\mathbf{r} \in \mathcal{L}_1} \cos\left[\sum_{I=1}^{3} t_m^I K_{mn} \Delta_I^2 \theta_n^{(1)}(\mathbf{r})\right] - \sum_{\mathbf{r} \in \mathcal{L}_2} \cos\left[\sum_{I=1}^{3} t_m^I K_{mn} \Delta_I^2 \theta_n^{(2)}(\mathbf{r})\right], \qquad (14)$$

where we have symmetrized the complex exponentials to render the Hamiltonian explicitly Hermitian.

We can write down an effective lattice action describing the ground state subspace of the Hamiltonian in Eq. (14). We achieve this by pinning the argument of the cosine to its minimum. This procedure leads to the action

$$S = \sum_{l=1,2} \int dt \sum_{\mathbf{r}} \left( \frac{1}{2\pi} K_{mn} \theta_m^{(l)}(\mathbf{r}, t) \partial_t \theta_n^{(l)}(\mathbf{r}, t) + \frac{1}{\pi} \theta_0^{(l)}(\mathbf{r}, t) t_m^I K_{mn} \Delta_I^2 \theta_n^{(l)}(\mathbf{r}, t) \right), \quad (15)$$

where $\theta_0^{(l)}$ can be regarded as a Lagrange multiplier implementing the ground state projection. Note that the lattice action has a well defined scaling if the microscopic fields $\theta_m^{(l)}$ are dimensionless in the standard sense, but the Lagrange multiplier $\theta_0^{(l)}$ must have dimension of inverse time. The action is invariant under the gauge transformation

$$\begin{aligned}
\theta_n^{(l)} &\to \theta_n^{(l)} + t_m^I K_{mn} \Delta_I^2 \zeta^{(l)}, \\
\theta_0^{(l)} &\to \theta_0^{(l)} + \partial_t \zeta^{(l)},
\end{aligned} \qquad (16)$$

with arbitrary dimensionless functions $\zeta^{(l)}(\mathbf{r}, t)$. To take the continuum limit, we define the rescaled fields

$$A_n^{(l)}(\mathbf{r}) = a_s^{-3/2} \theta_n^{(l)}(\mathbf{r}), \qquad\qquad A_0^{(l)}(\mathbf{r}) = a_s^{1/2} \theta_0^{(l)}(\mathbf{r}), \qquad (17)$$

where $a_s$ is the lattice spacing. Note that the dimension of $A_0^{(l)}$ has a contribution from the dimension of $\theta_0^{(l)}$. If we consider a short-time cutoff $a_t \sim a_s^{[t]}$ with dynamic exponent $[t] = 2$, then $A_0^{(l)}$ has the same spacetime dimension as $A_n^{(l)}$. We also replace $\frac{1}{a_s^2} \Delta_I^2 \to \partial_I^2$ and $a_s^3 \sum_{\mathbf{r}} \to \int d^3 r$. As a result, the continuum action reads

$$S = \sum_{l=1,2} \int d^3 r \, dt \left( \frac{1}{2\pi} K_{mn} A_m^{(l)} \partial_0 A_n^{(l)} + \frac{1}{\pi} A_0^{(l)} K_{mn} D_m A_n^{(l)} \right), \qquad (18)$$

with the derivative operators

$$D_m = \sum_{I=1}^{3} t_m^I \partial_I^2. \qquad (19)$$

Equation (18) is equivalent to the Chern-Simons-like theory proposed in Refs. [9, 20, 21] with an additional copy. The two copies with $l = 1, 2$ are related by the lattice translation $\mathbf{r} \mapsto \mathbf{r} + a_s(1, 1, 1)$, which exchanges the sublattices AB $\leftrightarrow$ CD. This property is reminiscent of Wen's plaquette model on the square lattice [53], where two types of excitations are related by a lattice translation. We note that the lattice spacing does not completely disappear from the continuum theory. For instance, it shows up in the gauge transformation

$$\begin{aligned}
A_n^{(l)} &\to A_n^{(l)} + a_s^{1/2} D_n \zeta^{(l)}, \\
A_0^{(l)} &\to A_0^{(l)} + a_s^{1/2} \partial_t \zeta^{(l)}.
\end{aligned} \qquad (20)$$

This is a manifestation of the UV-IR mixing inherent to fracton theories. This microscopic parameter will eventually need to be restored to make sense of the gauge structure as well as some physical properties computed in the field theory.

To keep track of the spatial derivatives, we find it convenient to switch to the notation

$$D_1 = \partial_x^2 - \partial_z^2 \equiv D_{xz}, \qquad D_2 = \partial_y^2 - \partial_z^2 \equiv D_{yz}, \tag{21}$$

$$A_1^{(l)}(\mathbf{r}) \equiv A_{xz}^{(l)}(\mathbf{r}), \qquad A_2^{(l)}(\mathbf{r}) \equiv A_{yz}^{(l)}(\mathbf{r}). \tag{22}$$

We then define the magnetic fields

$$B^{(l)}(\mathbf{r}) = D_{xz} A_{yz}^{(l)}(\mathbf{r}) - D_{yz} A_{xz}^{(l)}(\mathbf{r}). \tag{23}$$

We can use the antisymmetric differential operator $D_{IJ} = \partial_I^2 - \partial_J^2$ for general values of the indices $I, J \in \{1, 2, 3\}$. Note that, in particular, $D_{xy} = D_{xz} - D_{yz}$. Along the same lines, we consider the antisymmetric field $A_{IJ}(\mathbf{r}) = -A_{JI}(\mathbf{r})$ with the $xy$ component defined as $A_{xy}(\mathbf{r}) \equiv A_{xz}(\mathbf{r}) - A_{yz}(\mathbf{r})$. With this convention, we define the electric fields as

$$E_{IJ}^{(l)}(\mathbf{r}) = \partial_t A_{IJ}^{(l)}(\mathbf{r}) - D_{IJ} A_0^{(l)}(\mathbf{r}), \tag{24}$$

so that $E_{xy}^{(l)}(\mathbf{r}) = E_{xz}^{(l)}(\mathbf{r}) - E_{yz}^{(l)}(\mathbf{r})$. Note that the electric and magnetic fields are gauge-invariant operators. We can then work with the action in Eq. (18) in the following equivalent form:

$$S = \frac{1}{2\pi} \sum_{l=1,2} \int d^3r \, dt \left( A_{xz}^{(l)} E_{yz}^{(l)} - A_{yz}^{(l)} E_{xz}^{(l)} + A_0^{(l)} B^{(l)} \right). \tag{25}$$

## 3.2 Symmetries and line operators

From the equation of motion of $A_{IJ}^{(l)}$, we can read from Eq. (25)

$$E_{JK}^{(l)} = \partial_t A_{JK}^{(l)} - D_{JK} A_0^{(l)} = 0. \tag{26}$$

Integrating Eq. (26) over the $JK$ plane, we obtain the conservation law

$$\frac{d}{dt} \int d\xi^I d\bar{\xi}^I A_{JK}^{(l)} = 0, \qquad (I \neq J \neq K), \tag{27}$$

where we introduced the coordinates $\xi^I = \frac{|\epsilon^{IJK}|}{\sqrt{2}}(x_J + x_K)$ and $\bar{\xi}^I = \frac{\epsilon^{IJK}}{\sqrt{2}}(x_J - x_K)$, corresponding to the directions of motion of dipoles in the lattice model. In this notation, we have $D_{JK} = 2\partial_{\xi^I}\partial_{\bar{\xi}^I}$. We refer to Eq. (27) as the electric symmetry, since it emerges from the vanishing electric field.

We can derive additional subsystem conserved quantities if we restrict the integration to be along lines instead of planes. Assuming that the theory is defined on a finite volume with periodic boundary conditions in all spatial directions, we obtain

$$\frac{d}{dt} \oint d\xi^I A_{JK}^{(l)} = 0, \quad \frac{d}{dt} \oint d\bar{\xi}^I A_{JK}^{(l)} = 0, \quad I \neq J \neq K, \tag{28}$$

where the integrations are along lines that wind around the system in the direction of $\xi^I$ and $\bar{\xi}^I$. There are also conserved quantities associated with ribbons of arbitrary width $w$:

$$\frac{d}{dt} \int_{\xi^I}^{\xi^I + w} d\xi^I \oint d\bar{\xi}^I A_{JK}^{(l)} = 0, \qquad \frac{d}{dt} \oint d\xi^I \int_{\bar{\xi}^I}^{\bar{\xi}^I + w} d\bar{\xi}^I A_{JK}^{(l)} = 0. \tag{29}$$

The generators of the above symmetry,

$$\exp\left[\frac{iq}{\sqrt{a_s}}\int_{\xi^I}^{\xi^I+w}d\xi^I\oint d\bar{\xi}^I A_{JK}^{(l)}\right], \quad \exp\left[\frac{iq}{\sqrt{a_s}}\oint d\xi^I\int_{\bar{\xi}^I}^{\bar{\xi}^I+w}d\bar{\xi}^I A_{JK}^{(l)}\right] \tag{30}$$

can be interpreted as dipoles with charges $\pm q$ separated by a distance $w$.

The model also exhibits a magnetic symmetry with a conserved charge given by the integral of the magnetic field over the entire system:

$$\frac{d}{dt}\int d^3r\, B^{(l)} = \int d^3r\left(D_{xz}A_{yz}^{(l)} - D_{yz}A_{xz}^{(l)}\right) = 0. \tag{31}$$

We can also define conserved quantities on manifolds of codimension one, of the form

$$\frac{d}{dt}\int du\,dv\, B^{(l)} = 0, \tag{32}$$

with $u \in \{\xi^x, \bar{\xi}^x\}$ and $v \in \{\xi^y, \bar{\xi}^y\}$.

## 4 Boundary theory

In this section we derive the the continuum description of the Chamon model with a boundary. We point out some analogies with $K$-matrix theory and discuss the constraints of subsystem symmetries on correlations of charged operators. We then analyze the effects of perturbations and how they relate to possible gapped and gapless boundary phases.

### 4.1 Boundary action and symmetries

The action in Eq. (25) is gauge invariant up to boundary terms. Let us consider the theory defined on a manifold $\mathcal{M} = \mathbb{R} \times \mathcal{V}$, with $\mathbb{R}$ representing the time direction and $\mathcal{V}$ the spatial volume with a boundary at $z = 0$, i.e., $\mathcal{V} = \mathbb{R}_x \times \mathbb{R}_y \times (-\infty, 0]$. The variation of the action under a general transformation of the fields reads

$$\delta S = \frac{1}{2\pi}\sum_{l=1,2}\int_{\partial\mathcal{M}}d^2r\,dt\left(\delta A_0^{(l)}\partial_z A_{xy}^{(l)} - A_{xy}^{(l)}\partial_z\delta A_0^{(l)} + A_0^{(l)}\partial_z\delta A_{xy}^{(l)} - \delta A_{xy}^{(l)}\partial_z A_0^{(l)}\right). \tag{33}$$

For the gauge transformation in Eq. (20), we obtain

$$\delta S = \frac{a_s^{1/2}}{2\pi}\sum_{l=1,2}\int_{\partial\mathcal{M}}d^2r\,dt\left(\partial_z\zeta^{(l)}E_{xy}^{(l)} - \zeta^{(l)}\partial_z E_{xy}^{(l)}\right). \tag{34}$$

As in standard Chern-Simons theories, gauge invariance may be restored by restricting the gauge transformations in the presence of a boundary. In this process, the gauge fields become physical degrees of freedom at the boundary. One possibility is to impose $A_0^{(l)}|_{\partial\mathcal{V}} = \partial_z A_0^{(l)}|_{\partial\mathcal{V}} = 0$, so that the variations $\delta A_0^{(l)}$ and $\partial_z\delta A_0^{(l)}$ vanish as well. A more general gauge-fixing condition that yields $\delta S = 0$ is

$$A_0^{(l)} = \kappa_{ll'}A_{xy}^{(l')}, \tag{35}$$

with the convention of summing over the repeated index $l'$. Here we allow for a linear combination of the fields with real coefficients $\kappa_{ll'}$ obeying $\kappa_{l'l} = \kappa_{ll'}$.

We can satisfy the zero-flux condition $B^{(l)} = 0$ in the ground state sector by writing the gauge fields in the form

$$A_{IJ}^{(l)}(\mathbf{r}, t) = a_s^{1/2} D_{IJ} \varphi_l(\mathbf{r}, t), \tag{36}$$

where $\varphi_l$ are dimensionless scalar fields to be associated with the boundary degrees of freedom. Substituting Eq. (36) into the bulk action Eq. (25), we find that the integrand is a total derivative that integrates to the boundary action

$$S_{\text{bd}} = \frac{a_s}{2\pi} \int_{\partial\mathcal{M}} d^2r \, dt \left( \partial_z \varphi_l \, \partial_t D_{xy} \varphi_l - \varphi_l \, \partial_t D_{xy} \partial_z \varphi_l \right) \tag{37}$$

independently of the choice of coefficients $\kappa_{ll'}$ in Eq. (35). Note that the boundary action involves the derivative of the scalar fields with respect to the direction perpendicular to the boundary. This derivative is related to the difference between fields in adjacent layers. To match the correct number of degrees of freedom between bulk and boundary, we demand that the derivative be a linear combination of the fields:

$$a_s \partial_z \varphi_I(x, y) \to c_{ll'} \varphi_{l'}, \tag{38}$$

with real coefficients $c_{ll'}$. These coefficients may depend on microscopic details of the boundary interactions in the generic theory, beyond the exactly solvable lattice model. Using integration by parts, we realize that only the antisymmetric part of $c_{ll'}$ contributes to the action, and we obtain

$$S_{\text{bd}} = \frac{1}{2\pi} \int_{\partial\mathcal{M}} d^2r \, dt \, (c_{l'l} - c_{ll'}) \varphi_l \partial_t D_{xy} \varphi_{l'}. \tag{39}$$

Assuming $c_{21} > c_{12}$, we can rescale the fields $\varphi_l \to (c_{21} - c_{12})^{-1/2} \varphi_l$ to cast the boundary action in the form

$$S_{\text{bd}} = \frac{1}{2\pi} \int_{\partial\mathcal{M}} d^2r \, dt \, K_{ll'} \varphi_l \partial_t D_{xy} \varphi_{l'}, \tag{40}$$

with the same $K$ matrix that appears in the bulk theory in Eq. (8). This action implies that upon quantization the pair of boundary fields must obey the equal-time commutation relation

$$\left[ \varphi_l(\mathbf{r}), D_{xy} \varphi_{l'}(\mathbf{r}') \right] = i\pi (K^{-1})_{ll'} \delta(\mathbf{r} - \mathbf{r}'). \tag{41}$$

We stress that the number of independent fields in the boundary theory is associated with two sets of planes in the bulk theory. Had we not labeled the two gauge fields from the start, we would have to drop the index $l$ in Eq. (37), but we would be forced to treat $\partial_z \varphi$ as independent from $\varphi$. In any case, we would reach the same conclusion about the doubling of the modes and would end up with a boundary action formally equivalent to Eq. (40). We will return to the interpretation of Eq. (38) in Sec. 4.3.

The action in Eq. (40) does not give rise to any dynamics since the corresponding Hamiltonian vanishes identically. This result is analogous to the derivation of the action for chiral edge modes from the Chern-Simons theory for quantum Hall states [54,55]. To determine the boundary dynamics, we note that the action is invariant under the shift symmetries

$$\varphi_l \to \varphi_l + f_l(\xi^z) + \bar{f}_l\left(\bar{\xi}^z\right), \tag{42}$$

where $f_l(\xi^z)$ and $\bar{f}_l(\bar{\xi}^z)$ are arbitrary functions of $\xi^z = \frac{1}{\sqrt{2}}(x+y)$ and $\bar{\xi}^z = \frac{1}{\sqrt{2}}(x-y)$. The usual global U(1) symmetry corresponds to choosing $f_l$ and $\bar{f}_l$ to be constants, but the general form of Eq. (42) is connected with the subsystem symmetries of the fractonic theory. We

then add to the action a term that only involves spatial derivatives and respects the above symmetries:

$$S_{\text{bd}} = \frac{1}{2\pi} \int_{\partial \mathcal{M}} d^2r \, dt \left( K_{ll'} \varphi_l \partial_t D_{xy} \varphi_{l'} - M_{ll'} D_{xy} \varphi_l D_{xy} \varphi_{l'} \right), \tag{43}$$

where $M$ is a symmetric matrix, as imposed by the even number of spatial derivatives in the last term. The shift symmetries are generated by the currents

$$J_{l,0} = \frac{1}{\pi} K_{ll'} D_{xy} \varphi_{l'}, \qquad J_{l,xy} = -\frac{1}{\pi} K_{ll'} \partial_t \varphi_{l'} - \frac{1}{\pi} M_{ll'} D_{xy} \varphi_{l'}, \tag{44}$$

which obey the continuity equations

$$\partial_0 J_{l,0} - D_{xy} J_{l,xy} = 0. \tag{45}$$

The latter are simply the equations of motion derived from the action in Eq. (43). Solving these equations in terms of Fourier modes with momentum $\mathbf{p} = (p_x, p_y)$, we find the dispersion relation

$$\omega^2 = \mu^2 \left( p_x^2 - p_y^2 \right)^2, \tag{46}$$

where we define

$$\mu = \sqrt{\det M}. \tag{47}$$

Thus, the stability of the boundary theory requires $\mu > 0$. In analogy with the velocity matrix of chiral edge states in the quantum Hall effect [54], hereafter we assume that $M$ is a positive-definite matrix. Note that $\mu$ is the single parameter governing the dispersion of the elementary excitations. It is also interesting to note that, using $\Phi = (\varphi_1, \varphi_2)$, we can define a duality transformation $\Phi \to \sigma^1 \Phi$, where $\sigma^1$ is the Pauli matrix acting on the two-component scalar field. This transformation maps $K \to -K$ and $M \to \sigma^1 M \sigma^1$ without affecting the dispersion parameter $\mu$. The sign change of the $K$ matrix preserves the algebra of the physical operators, and can be used to implement the proper re-scaling of the fields above Eq. (40) in case $c_{21} < c_{12}$.

In addition to the conserved currents in Eq. (44), the boundary action has a dipole winding symmetry generated by the currents

$$\mathcal{J}_{l,0} = \frac{1}{2\pi} \partial_{\xi^z} \partial_{\bar{\xi}^z} \varphi_l, \qquad \mathcal{J}_{l,\xi^z \bar{\xi}^z} = \frac{1}{2\pi} \partial_t \varphi_l, \tag{48}$$

obeying $\partial_t \mathcal{J}_{l,0} = \partial_{\xi^z} \partial_{\bar{\xi}^z} \mathcal{J}_{l,\xi^z \bar{\xi}^z}$. The corresponding charges of the winding symmetry are dipole configurations that wind around the system in either $\xi^z$ or $\bar{\xi}^z$ directions:

$$\frac{1}{2\pi} \int_{\xi^z}^{\xi^z+w} d\xi^z \oint d\bar{\xi}^z \, \partial_{\xi^z} \partial_{\bar{\xi}^z} \varphi_l, \qquad \frac{1}{2\pi} \oint d\xi^z \int_{\bar{\xi}^z}^{\bar{\xi}^z+w} d\bar{\xi}^z \, \partial_{\xi^z} \partial_{\bar{\xi}^z} \varphi_l. \tag{49}$$

The latter are reminiscent of the bulk theory, cf. Eq. (30).

The boundary Hamiltonian derived from Eq. (43) reads

$$H_{\text{bd}} = \frac{1}{2\pi} \int_{\partial \mathcal{V}} d^2r \, M_{ll'} D_{xy} \varphi_l D_{xy} \varphi_{l'}. \tag{50}$$

To diagonalize the Hamiltonian, we assume periodic boundary conditions in the $xy$ plane and employ the mode expansion

$$\varphi_l(\mathbf{r}) = \frac{1}{\sqrt{L\bar{L}}} \sum_{n,\bar{n} \in \mathbb{Z}} \varphi_{l,n,\bar{n}} \, e^{ip_n \xi^z + i\bar{p}_{\bar{n}} \bar{\xi}^z}, \tag{51}$$

where $L$ and $\bar{L}$ are the lengths in the directions of $\xi^z$ and $\bar{\xi}^z$, respectively, and $p_n = 2\pi n/L$ and $\bar{p}_{\bar{n}} = 2\pi \bar{n}/\bar{L}$ are the discrete momenta. Defining $\rho_{l,n,\bar{n}} = 2 p_n \bar{p}_{\bar{n}} \varphi_{l,n,\bar{n}}$, we find that the normal-mode operators obey a generalized U(1) Kac-Moody algebra

$$\left[ \varphi_{l,n,\bar{n}}, \rho_{l',n',\bar{n}'} \right] = i\pi \left( K^{-1} \right)_{ll'} \delta_{n,-n'} \delta_{\bar{n},-\bar{n}'} . \tag{52}$$

In terms of $\rho_{l,n,\bar{n}}$, the Hamiltonian becomes

$$H_{\mathrm{bd}} = \frac{1}{2\pi} \sum_{n,\bar{n} \neq 0} M_{ll'} \rho_{l,n,\bar{n}} \rho_{l',-n,-\bar{n}} . \tag{53}$$

Using a Bogoliubov transformation (see Appendix A for details), we obtain

$$H_{\mathrm{bd}} = \sum_{n,\bar{n} \neq 0} \omega_{n,\bar{n}} b^{\dagger}_{n,\bar{n}} b_{n,\bar{n}} + \mathrm{const.}\,, \tag{54}$$

where $b^{\dagger}_{n,\bar{n}}$ and $b_{n,\bar{n}}$ are bosonic creation and annihilation operators, respectively, associated with momentum $\mathbf{p} = \left( \frac{1}{\sqrt{2}}(p_n + \bar{p}_{\bar{n}}), \frac{1}{\sqrt{2}}(p_n - \bar{p}_{\bar{n}}) \right)$ and energy $\omega_{n,\bar{n}} = 2\mu |p_n \bar{p}_{\bar{n}}|$.

In the thermodynamic limit $L, \bar{L} \to \infty$, the Hamiltonian in Eq. (50) predicts gapless boundary modes with an anisotropic quadratic dispersion. In fact, the excitation energy vanishes along the lines $p_y = \pm p_x$ in momentum space for arbitrary values of $|\mathbf{p}|$. As a consequence, short-wavelength modes can contribute to the low-energy physics, another hallmark of the UV-IR mixing. However, we still need to analyze the effect of symmetry-allowed interactions that may gap out the boundary spectrum, similarly to what happens to edge modes of topological phases without charge-conservation symmetries [39–41]. To understand this point, we must first identify the operators that create charged excitations at the boundary.

## 4.2 Charged operators

From Eq. (44), we can define the charge operators

$$Q_l = \frac{1}{\pi} \int_{\partial \mathcal{V}} d^2 r \, K_{ll'} D_{xy} \varphi_{l'} . \tag{55}$$

The operators charged under $Q_l$ are the exponentials

$$\Psi_q(\mathbf{r}) \sim e^{i q_l \varphi_l(\mathbf{r})}, \tag{56}$$

where we label the fields by the charge vector $q = (q_1, q_2)$, so that $\Psi_{-q}(\mathbf{r}) = \Psi^{\dagger}_q(\mathbf{r})$. Using Eq. (41), we obtain

$$\left[ Q_l, \Psi_q(\mathbf{r}) \right] = -q_l \Psi_q(\mathbf{r}) . \tag{57}$$

Expressing the position $\mathbf{r} \in \partial \mathcal{V}$ in terms of the coordinates $\xi^z$ and $\bar{\xi}^z$, we can write the commutation relation for the $\varphi_l$ fields as

$$\left[ \varphi_l \left( \xi^z_1, \bar{\xi}^z_1 \right), \varphi_{l'} \left( \xi^z_2, \bar{\xi}^z_2 \right) \right] = \frac{i\pi}{8} \left( K^{-1} \right)_{ll'} \mathrm{sgn} \left( \xi^z_1 - \xi^z_2 \right) \mathrm{sgn} \left( \bar{\xi}^z_1 - \bar{\xi}^z_2 \right) , \tag{58}$$

where $\mathrm{sgn}(x)$ is the sign function. As a consequence, the charged operators obey the algebra

$$\Psi_q \left( \xi^z_1, \bar{\xi}^z_1 \right) \Psi_{q'} \left( \xi^z_2, \bar{\xi}^z_2 \right) = e^{-\frac{i\pi}{8} q_l q_{l'} \left( K^{-1} \right)_{ll'} \mathrm{sgn}(\xi^z_1 - \xi^z_2) \mathrm{sgn}(\bar{\xi}^z_1 - \bar{\xi}^z_2)} \Psi_{q'} \left( \xi^z_2, \bar{\xi}^z_2 \right) \Psi_q \left( \xi^z_1, \bar{\xi}^z_1 \right) . \tag{59}$$

In particular, the operators with "elementary" charges $q = (1,0)$ and $q' = (0,1)$ do not commute with each other. We shall think of $\Psi^{\dagger}_{(1,0)}$ and $\Psi^{\dagger}_{(0,1)}$ as creating monopoles at the boundary

of the system. We note that from the field theory alone it is not entirely clear how the charges and the entries of the $K$ matrix should be quantized, since the lattice spacing $a_s$ appears in the gauge structure of the model. Here we rely on information from the lattice model, where the elements of the $K$ matrix are quantized according to Eq. (10), and impose an analogous quantization condition for the boundary fields. Remarkably, our construction suggests a generalization to fracton models described by higher-level $K$ matrices [20], which can harbor excitations with fractional charge. Here we focus on the model described by the $K$ matrix in Eq. (8) corresponding to the $\mathbb{Z}_2$ Chamon model.

The symmetries of the model strongly constrain the correlations of charged operators. Let us consider $n$-point functions of the form

$$\left\langle \prod_\alpha \Psi_{q^\alpha}(\mathbf{r}_\alpha) \right\rangle \sim \left\langle \prod_\alpha e^{iq_l^\alpha \varphi_l(\mathbf{r}_\alpha)} \right\rangle . \tag{60}$$

The shift symmetries in Eq. (42) impose that the correlations are non-vanishing only if

$$\sum_\alpha q_l^\alpha \left[ f_l\left(\xi_\alpha^z\right) + \bar{f}_l\left(\bar{\xi}_\alpha^z\right) \right] = 0 , \tag{61}$$

with arbitrary functions $f_l\left(\xi_\alpha^z\right)$ and $\bar{f}_l\left(\bar{\xi}_\alpha^z\right)$. Beyond the usual charge neutrality condition, Eq. (61) implies that the charge of an excitation created by $\Psi_{q^\alpha}(\mathbf{r}_\alpha)$ is effectively position dependent [56,57]. This condition gives us information about the mobility of excitations at the boundary. The immobility of fractons is manifested in the vanishing of the correlation for a set of isolated positions $\{\mathbf{r}_\alpha\}$. On the other hand, correlations of dipoles made up of opposite charges separated by a short distance $w \sim a_s$ in the $\xi^z$ direction have the form

$$\left\langle \prod_\alpha e^{iq_l^\alpha \varphi_l(\xi_\alpha^z+w,\bar{\xi}_\alpha^z)} e^{-iq_l^\alpha \varphi_l(\xi_\alpha^z,\bar{\xi}_\alpha^z)} \right\rangle \sim \left\langle \prod_\alpha e^{iwq_l^\alpha \partial_{\xi^z} \varphi_l(\xi_\alpha^z,\bar{\xi}_\alpha^z)} \right\rangle . \tag{62}$$

These correlations have to obey a less restricted rule

$$\sum_\alpha q_l^\alpha f_l'\left(\xi_\alpha^z\right) = 0 , \tag{63}$$

where $f_l'(x)$ is the derivative of $f_l(x)$. In this case, we obtain a nonzero correlation for two dipoles at positions corresponding to the same value of $\xi^z$ but arbitrary values of $\bar{\xi}^z$, as expected because these dipoles can propagate along lines in the $\bar{\xi}^z$ direction. Similarly, dipoles composed of charges separated by $w \sim a_s$ in the $\bar{\xi}^z$ direction are correlated along the $\xi^z$ direction.

## 4.3 Boundary line operators

The line operators described in Sec. 3.2 must be modified when one of their endpoints lies at the boundary. Analyzing these operators, we would like to verify that the effective field theory is able to recover the boundary processes discussed for the lattice model in Sec. 2.

Consider a dipole with charges separated by a small distance $w \sim a_s$ along the $\bar{\xi}^x$ direction in a $yz$ plane; cf. the lattice representation in Fig. 5. We bring the dipole from infinity to the boundary on the $z = 0$ plane along a line $\Gamma$ in the positive $\xi^x$ direction. Let $\mathbf{r}_0 = (x_0, y_0, 0)$ be the endpoint of $\Gamma$ on the boundary. The motion of the dipole is described by the operator

$$W_l = \exp\left( \frac{i}{\sqrt{a_s}} \int_{\bar{\xi}_0^x}^{\bar{\xi}_0^x+w} d\bar{\xi}^x \int_{-\infty}^{\xi_0^x} d\xi^x A_{yz}^{(l)} \right) , \tag{64}$$

where $\xi_0^x = \bar{\xi}_0^x = y_0/\sqrt{2}$. Using Eq. (36) and $D_{yz} = 2\partial_{\xi^x}\partial_{\bar{\xi}^x}$, we can perform the integrals and express the line operators in terms of the $\varphi_l$ fields as

$$W_l = \exp\left[2i\varphi_l(\mathbf{r}_0') - 2i\varphi_l(\mathbf{r}_0)\right], \tag{65}$$

where $\mathbf{r}_0' = \mathbf{r}_0 + \frac{1}{\sqrt{2}}(0, w, -w)$. Expanding for small $w$, we obtain

$$W_l = \exp\left[i\sqrt{2}w\partial_y\varphi_l(\mathbf{r}_0) - i\sqrt{2}w\partial_z\varphi_l(\mathbf{r}_0)\right]. \tag{66}$$

Using Eq. (38) to eliminate the derivative in the $z$ direction, we obtain an expression well defined within the boundary theory:

$$W_l = \exp\left[-i\frac{\sqrt{2}w}{a_s}c_{ll'}\varphi_{l'}(\mathbf{r}_0) + i\sqrt{2}w\partial_y\varphi_l(\mathbf{r}_0)\right]. \tag{67}$$

Comparing the above expression with the charged operator in Eq. (56), we recognize that the factor $e^{-i\frac{\sqrt{2}w}{a_s}c_{ll'}\varphi_{l'}}$ creates a fracton with charge $q = \frac{\sqrt{2}w}{a_s}(c_{l1}, c_{l2})$. Charge quantization imposes $(\sqrt{2}w/a_s)c_{ll'} \in \mathbb{Z}$. This result provides further interpretation for the boundary condition in Eq. (38): The coefficients $c_{ll'}$ encode the fusion of a dipoles into a boundary fracton at the endpoint of the line operator. Recall that there are two types of boundaries, distinguished by which sublattice parities are broken and by the types of fractons that can appear at the termination of BHCs. For instance, if dipoles can be converted into fractons with charge $q = (1, 0)$, but not with charge $q = (0, 1)$, we must have $c_{l2} = 0$. Moreover, the factor $e^{i\sqrt{2}w\partial_y\varphi_l(\mathbf{r}_0)}$ in Eq. (67) can be interpreted in terms of the creation of additional dipoles that propagate along the boundary. Note that we can decompose $\partial_y\varphi_l = \frac{1}{\sqrt{2}}\partial_{\xi^z}\varphi_l - \frac{1}{\sqrt{2}}\partial_{\bar{\xi}^z}\varphi_l$, where the derivatives with respect to $\xi^z$ and $\bar{\xi}^z$ correspond to the natural orientation of the dipoles in the boundary plane. Similar combinations appear when we consider operators associated with dipoles that propagate to the boundary along a line in an $xz$ plane.

## 4.4 Perturbations and gapping conditions

The lattice model discussed in Sec. 2 has gapped boundary modes since, like in the bulk, defects of the boundary stabilizers cost finite energy. To recover this result within the continuum theory, we must consider perturbations that break the U(1) symmetries of the Hamiltonian in Eq. (50) by creating charged excitations at the boundary. Defining such gapping terms can also be motivated by an analogy with the standard theory of $K$ matrix Chern-Simons. In (2+1) dimensions, a bulk $\mathcal{M}$ described by an odd number of copies of Chern-Simons theories will always have protected gapless modes at the edge, $\partial\mathcal{M}$, which are described by copies of chiral boson theories [54]. This is a consequence of the gravitational anomaly (non-zero thermal Hall conductance) [58] that is standard in such theories when defined in a manifold with a boundary. For such theories the presence of the gravitational anomaly makes it impossible to fully gap the edge modes. This is no longer the case when the bulk is described by an even number of Chern-Simons theories. In this situation there can be anomaly cancelation, due to the even number of chiral bosons at the edge, resulting in a vanishing thermal Hall conductance. Therefore, there is no obvious obstruction to gapping the edge modes. In fact, these edge theories can be gapped given that the gapping terms obey a list of criterias [39, 41].

Inspired by the lower dimensional case we follow a similar route. The boundary theory (50) is a generalization of the standard chiral boson theories. The even number of fields at the boundary suggests that there can be an anomaly cancelation and therefore one can look for appropriate interactions that fully gap the boundary modes. These gapping terms are not arbitrary and must obey some criterias in order for the gapped spectrum to be stable, in analogy to the standard cases [39, 41].

We start with a perturbation of the form

$$\delta H_q = -g_q \int_{\partial \mathcal{V}} d^2r \, \cos(q_l \varphi_l), \tag{68}$$

with $g_q > 0$. This interaction corresponds to a process that creates or annihilates defects with charge $q = (q_1, q_2)$. If we assume that the charges are quantized with $q_l \in \mathbb{Z}$, the interaction breaks the continuous symmetries in Eq. (42) down to the discrete subsystem symmetries

$$\varphi_l\left(\xi^z, \bar{\xi}^z\right) \to \varphi_l\left(\xi^z, \bar{\xi}^z\right) + 2\pi m_l\left(\xi^z\right) + 2\pi \bar{m}_l\left(\bar{\xi}^z\right), \tag{69}$$

with $q_l m_l(\xi^z), q_l \bar{m}_l(\bar{\xi}^z) \in \mathbb{Z}$.

Let us analyze the effects of $\delta H_q$ in the strong coupling limit $g_q \to \infty$. In a semiclassical picture, the cosine potential pins the fields to one of its minima. The expansion of the cosine around $\varphi_l = 0$ generates a mass term $\delta H_q \sim \frac{1}{2} g_q (q_l \varphi_l)^2$. For small momentum, we obtain the dispersion

$$\omega^2 \approx \Delta^2 + \mu^2 (p_x^2 - p_y^2)^2, \tag{70}$$

with a gap given by $\Delta = (g_q M_{ll'} q_l q_{l'})^{1/2}$. Note that $\Delta > 0$ for any $q$ since $M$ is positive-definite.

More generally, the boundary action may contain multiple cosine terms with different charges. It is possible to consistently minimize two cosine potentials $\delta H_q$ and $\delta H_{q'}$ if they commute with one another, which is ensured by the "null statistics" condition [see Eq. (59)]

$$q_l (K^{-1})_{ll'} q_l' \in 16\mathbb{Z}. \tag{71}$$

For $q' = q$, this condition is trivially satisfied because $K^{-1}$ is antisymmetric. Interaction terms with $q' \neq q$ that obey Eq. (71) are called compatible [41]. In this case, the interactions generate a stable gap in the dispersion as in Eq. (70). Otherwise, if the interactions are incompatible, the attempt to pin the potentials simultaneously gives rise to singular terms such as $\omega^2 \sim (p_x^2 - p_y^2)^{-1}$, which invalidates the expansion of the cosines.

For the Chamon model, the null statistics condition implies that the cosine terms associated with elementary charges $q = (1, 0)$ and $q = (0, 1)$ are incompatible. To generate a gap, we can pin either $\varphi_1 = 0$ or $\varphi_2 = 0$, depending on which perturbation has a larger coupling constant. We can interpret this result in terms of the two types of (001) boundaries, distinguished by the types of fractons with broken parity constraints. These gapping conditions can be straightforwardly generalized to boundary theories where the scalar field $\varphi_l$ contains more than two components [20].

So far we have considered the limit of large $g_q$, but another important question concerns the relevance of the boundary interactions at weak coupling. Here we should note that the action in Eq. (43) is scale invariant with dynamic exponent $[t] = 2$. However, unlike the quantum Lifshitz theory in $2 + 1$ dimensions [59, 60], our fractonic theory lacks continuous rotation invariance in the boundary plane and the vanishing of the dispersion along lines in momentum space poses challenges for a standard renormalization group analysis [61]. Nevertheless, we may explore a possible effective scaling dimension of the operator in Eq. (68) by calculating its correlation function in the unperturbed model. We proceed in analogy with the computation of correlators of vertex operators in conformal field theory [62]. We find that the equal-time correlation of charged operators vanishes exactly (see Appendix B):

$$\left\langle e^{iq_l \varphi_l(\mathbf{r})} e^{-iq_{l'} \varphi_{l'}(\mathbf{0})} \right\rangle = 0, \qquad (\mathbf{r} \neq 0). \tag{72}$$

This result can be traced back to the subsystem symmetries and the generalized neutrality condition in Eq. (61), which is not satisfied for any $\mathbf{r} \neq 0$. Thus, the correlation of charged

operators is effectively short-ranged in all spatial directions. Similar behavior appears in effective theories for the Bose metal phase in two dimensions [63] and the classical plaquette-dimer model in three dimensions [61, 64]. The short-range correlations indicate that $\delta H_q$ is irrelevant at weak coupling. This conclusion is supported by a perturbative renormalization group analysis [64], adapted to incorporate the anisotropic dispersion and the large-momentum contributions to the low-energy physics.

However, the discussion in Section 4.2 suggests that operators that create dipoles can have non-vanishing correlations. For example, consider the perturbation

$$\delta H_q' = -g_q' \int_{\partial\mathcal{V}} d^2r \, \cos\left(wq_l \partial_{\xi^z} \varphi_l\right) - g_q' \int_{\partial\mathcal{V}} d^2r \, \cos\left(wq_l \partial_{\bar{\xi}^z} \varphi_l\right), \tag{73}$$

where we set the same coupling constant for both terms assuming that the boundary preserves the $C_4$ rotation symmetry that takes $\xi^z \mapsto \bar{\xi}^z$, $\bar{\xi}^z \mapsto -\xi^z$. Taking two points along a line with fixed $\xi^z$, we obtain the correlation (see Appendix B)

$$\left\langle e^{iwq_l \partial_{\xi^z} \varphi_l(\xi^z=0,\bar{\xi}^z)} e^{-iwq_{l'} \partial_{\xi^z} \varphi_{l'}(0,0)} \right\rangle = \left[ \frac{1}{\alpha_s^2 + \left(\bar{\xi}^z\right)^2} \right]^{\eta_q}, \tag{74}$$

where $\alpha_s \sim a_s$ is a short-distance cutoff and the exponent is given by

$$\eta_q = \frac{(w/\alpha_s)^2}{8\pi\mu} q_l M_{ll'} q_{l'}. \tag{75}$$

On general grounds, we expect a relevant operator to be associated with slowly decaying correlations, i.e., a small value of $\eta_q$ [61]. This reasoning can be justified more rigorously by an RG analysis as discussed in Ref. [64]. Note that the exponent $\eta_q$ is tied to the ratio between non-universal microscopic parameters. Nevertheless, we can make some qualitative considerations assuming that $\alpha_s$ is of the same order as the dipole length $w$.

Clearly, the effective scaling dimension of the dipole operator increases with the charges $q_l \in \mathbb{Z}$. For the operators with charges $(n,0)$ and $(0,n)$, we have

$$\eta_{(n,0)} = \frac{n^2(w/\alpha_s)^2 M_{11}}{8\pi\mu}, \qquad \eta_{(0,n)} = \frac{n^2(w/\alpha_s)^2 M_{22}}{8\pi\mu}. \tag{76}$$

Similarly to the quantum Lifshitz theory [60], the relevance of the perturbation depends on parameters that govern the dispersion relation of the excitations, see Eq. (46). To understand the dependence on the matrix elements of $M$, let us first discuss the case where $M$ is diagonal. In this case, we have $\mu = \sqrt{M_{11}M_{22}}$. If we fix $w/\alpha_s \approx 1$, the exponents in Eq. (76) only depend on the ratio $\sqrt{M_{11}/M_{22}}$. Decreasing $\sqrt{M_{11}/M_{22}}$ increases $\eta_{(0,n)}$, while making $\eta_{(n,0)}$ smaller. Thus, we cannot make both exponents arbitrarily large at the same time. We take this as a sign that, if $M$ is diagonal in the "flavor" basis in which dipoles are created, the theory contains at least one relevant operator at weak coupling. In contrast, for non-diagonal $M$ we have $M_{12}$ as an additional parameter. In this case, for fixed $M_{11}$ and $M_{22}$, we can get $\eta_q \to \infty$ for all charges by taking $|M_{12}| \to \sqrt{M_{11}M_{22}}$, so that $\mu \to 0$. In this regime, all dipole perturbations become irrelevant, regardless of the choice of $w/\alpha_s$. Note that this condition can happen close to the edge of stability of the boundary theory, since the spectrum would become imaginary for $|M_{12}| > \sqrt{M_{11}M_{22}}$.

Our effective field theory then suggests the existence of a stable *gapless* boundary phase at weak coupling and for sufficiently small $\mu$. Provided that all perturbations are irrelevant, the low-energy boundary modes exhibit emergent continuous subsystem symmetries as described

by the quadratic Hamiltonian in Eq. (50). Even though we only encountered gapped boundaries in the analysis of the lattice model in Sec. 2, the physical conditions for finding gapless boundary modes can be illuminated by an analogy with the quantum Hall effect. From the perspective of a hydrodynamic approach [54], the velocity of the chiral edge mode in a quantum Hall fluid is proportional to the electric field of the confining potential near the edge. As a consequence, electrons propagate with a lower velocity when the confining potential varies smoothly in the direction perpendicular to the edge. Along the same lines, we expect that in our theory the dispersion parameter $\mu$ could be smaller for a smoothly varying boundary between the fractonic phase and a trivial phase. The spatial dependence can be controlled by considering, for instance, the model defined in infinite volume including a Zeeman term $H_Z = -\sum_{\mathbf{r}} \mathbf{h}(\mathbf{r}) \cdot \boldsymbol{\sigma}_{\mathbf{r}}$ with an inhomogeneous magnetic field $\mathbf{h}(\mathbf{r}) = \lambda(z-z_0)^2 \Theta(z-z_0)\hat{\mathbf{z}}$, where $\Theta(x)$ is the Heaviside step function. The solvable lattice model of Sec. 2 corresponds to a sharp boundary in the limit $\lambda \to \infty$. In contrast, a smooth boundary characterized by a small value of $\lambda$ introduces a small energy scale in the problem and should give rise to slower boundary modes. Alternatively, one may try to tune the dispersion parameter $\mu$ by adding suitable boundary perturbations on a sharp boundary so as to enhance the flavor mixing contained in the off-diagonal matrix element $M_{12}$.

Starting from the fixed point of gapless boundary modes, the system can undergo a fractonic Berezinskii-Kosterlitz-Thouless (BKT) transition when the dipole operators become relevant [61, 64]. This transition is driven by the proliferation of dipole excitations. If we assume that $g'_q$ flows to strong coupling and pin the cosine potential in Eq. (73), the expansion around the minimum generates terms proportional to $\left(\partial_{\xi^z}\varphi_l\right)^2$ and $\left(\partial_{\bar{\xi}^z}\varphi_l\right)^2$. These terms break the subsystem symmetries in Eq. (42), destroying the fracton physics at the boundary and allowing the excitations to recover full mobility. It is worth mentioning that a similar transition in 2+1 dimensions has been discussed in the context of the Xu-Moore model for $p + ip$ superconducting arrays [65, 66]. In the latter, it has been argued that the subsystem symmetries imply a dimensional reduction in the effective theory for the transition. Note that we can in principle push the boundary of the Chamon model towards the BKT-type transition by decreasing the parameter $M_{12}$. On the other hand, the gapless phase is also destabilized if $|M_{12}|$ is large enough so that the dispersion parameter $\mu$ becomes imaginary. Therefore, we expect the gapless phase to appear as an intermediate phase as a function of a control parameter that tunes $M_{12}$. We leave a detailed study of boundary phase transitions in the Chamon model for future work.

## 5 Discussion

We have examined the boundary theory of the Chamon model from the lattice and continuum perspectives. We focused our attention on a specific type of boundary on a (001) plane where five-site boundary stabilizers violate two out of four parity constraints. The effective field theory for the boundary was formulated in terms of a two-component bosonic field and resembles the usual $K$-matrix Chern-Simons descriptions. While our boundary action is compatible with a recent study of the X-cube model [45], the boundary condition in the Chamon model involves the normal derivative of the boundary field. Physically, this boundary condition represents processes that convert dipoles into fractons. We believe that our formulation can be useful to describe other fracton boundaries, for example boundaries of the X-cube model along planes other than the (001) plane.

We also analyzed the effects of perturbations to the quadratic boundary Hamiltonian. At strong coupling, we found that charged operators can open a gap in the boundary spectrum. The theory allows for (at least) two types of gapped boundaries, obtained by pinning either

component of the boundary field. We expect the exactly solvable lattice model with a sharp boundary to be in the strong-coupling regime since the model does not contain any small parameter. On the other hand, the weak-coupling regime might be realized in a generic model where perturbations introduce quantum fluctuations and the boundary can be smooth on the scale of the lattice parameter. Analyzing the weak-coupling regime, we found that the charged operators that create monopoles are always irrelevant because their correlations are short ranged in all spatial directions. In this regime, the leading perturbation is related to the creation of dipoles, and it can also be irrelevant depending on the model parameters. As a result, the effective field theory suggests the existence of a stable gapless boundary phase which is beyond the lattice model with a sharp boundary.

Our results raise several questions for future work. One question is whether other types of boundaries of fracton phases can also be described by a generalization of the $K$-matrix theory discussed here. For instance, it would be interesting to investigate $\mathbb{Z}_n$ models that may harbor fractional excitations at the boundary, in analogy with the chiral edge modes of fractional quantum Hall fluids. In addition, the boundary processes are geometry dependent, and different boundaries can be obtained by simply changing the direction of the surface plane. A more ambitious goal would be to map out boundary phase diagrams of fractonic models. For topological phases in 2+1 dimensions, it is possible to tune boundary interactions to drive phase transitions without closing the bulk gap, and the universality classes are described by conformal field theories familiar from symmetry-breaking transitions in 1+1 dimensions [42]. One may then wonder whether some exotic 2+1 critical theories might be realized at the boundary of fractonic systems.

# Acknowledgements

We thank Claudio Chamon for helpful discussions.

# Funding information

This work was supported by a grant from the Simons Foundation (1023171, W.B.F., R.G.P.). W.B.F. acknowledges FUNPEC under grant 182022/1707. R.G.P. acknowledges funding by Brazilian agency CNPq. Research at IIP-UFRN is supported by Brazilian ministries MEC and MCTI.

# A Diagonalization of the boundary Hamiltonian

In this appendix we discuss the details about the diagonalization of the Hamiltonian in Eq. (53). The mode expansion for the fields can be written as

$$\varphi_l(\mathbf{r}) = \frac{1}{\sqrt{L\bar{L}}} \sum_{\mathbf{n} \neq 0} \varphi_{l,\mathbf{n}} e^{i\mathbf{p_n} \cdot \mathbf{r}}, \tag{A.1}$$

where $\mathbf{n} = (n, \bar{n}) \in \mathbb{Z}^2$, $\mathbf{p_n} = (p_n, \bar{p}_{\bar{n}}) = \left(2\pi n/L, 2\pi\bar{n}/\bar{L}\right)$, and $\varphi_{l,-\mathbf{n}} = \varphi_{l,\mathbf{n}}^\dagger$. Using $\rho_{l,\mathbf{n}} = 2p_n\bar{p}_{\bar{n}}\varphi_{l,\mathbf{n}}$, we can write the Hamiltonian explicitly as

$$H_{\text{bd}} = \frac{2}{\pi} \sum_{\mathbf{n} \neq 0} p_n^2 \bar{p}_{\bar{n}}^2 \left(M_{11}\varphi_{1,\mathbf{n}}\varphi_{1,-\mathbf{n}} + M_{22}\varphi_{2,\mathbf{n}}\varphi_{2,-\mathbf{n}} + 2M_{12}\varphi_{1,\mathbf{n}}\varphi_{2,-\mathbf{n}}\right). \tag{A.2}$$

Next, we write

$$\varphi_{1,\mathbf{n}} = -\sqrt{\frac{\pi}{4|p_n \bar{p}_{\bar{n}}|}}\left(a_{\mathbf{n}} + a_{-\mathbf{n}}^{\dagger}\right), \tag{A.3}$$

$$\varphi_{2,\mathbf{n}} = -i\,\mathrm{sgn}(n\bar{n})\sqrt{\frac{\pi}{4|p_n \bar{p}_{\bar{n}}|}}\left(a_{\mathbf{n}} - a_{-\mathbf{n}}^{\dagger}\right), \tag{A.4}$$

where $a_{\mathbf{n}}$ and $a_{\mathbf{n}}^{\dagger}$ are annihilation and creation operators obeying $\left[a_{\mathbf{n}}, a_{\mathbf{m}}^{\dagger}\right] = \delta_{\mathbf{n},\mathbf{m}}$. In terms of $a_{\mathbf{n}}$ and $a_{\mathbf{n}}^{\dagger}$, the Hamiltonian in Eq. (A.2) becomes

$$H_{\mathrm{bd}} = \frac{1}{2}\sum_{\mathbf{n}}\left(\epsilon_{\mathbf{n}}a_{\mathbf{n}}^{\dagger}a_{\mathbf{n}} + \epsilon_{\mathbf{n}}a_{-\mathbf{n}}^{\dagger}a_{-\mathbf{n}} + \zeta_{\mathbf{n}}a_{\mathbf{n}}a_{-\mathbf{n}} + \zeta_{\mathbf{n}}^{*}a_{\mathbf{n}}^{\dagger}a_{-\mathbf{n}}^{\dagger}\right) + \mathrm{const.}, \tag{A.5}$$

where $\epsilon_{\mathbf{n}}$ and $\zeta_{\mathbf{n}}$ are defined as

$$\epsilon_{\mathbf{n}} = |p_n \bar{p}_{\bar{n}}|\left(M_{11} + M_{22}\right), \qquad \zeta_{\mathbf{n}} = |p_n \bar{p}_{\bar{n}}|\left[M_{11} - M_{22} + 2i\,\mathrm{sgn}(n\bar{n})M_{12}\right]. \tag{A.6}$$

The Hamiltonian can now be diagonalized by a Bogoliubov transformation. We define a new set of operators

$$b_{\mathbf{n}} = u_{\mathbf{n}}a_{\mathbf{n}} - v_{\mathbf{n}}a_{-\mathbf{n}}^{\dagger}, \qquad b_{\mathbf{n}}^{\dagger} = u_{\mathbf{n}}^{*}a_{\mathbf{n}}^{\dagger} - v_{\mathbf{n}}^{*}a_{-\mathbf{n}}, \tag{A.7}$$

with $u_{-\mathbf{n}} = u_{\mathbf{n}}$ and $v_{-\mathbf{n}} = v_{\mathbf{n}}$. The transformation preserves the commutation relations if the coefficients $u_{\mathbf{n}}$ and $v_{\mathbf{n}}$ satisfy $|u_{\mathbf{n}}|^2 - |v_{\mathbf{n}}|^2 = 1$. Choosing the coefficients so that

$$|u_{\mathbf{n}}|^2 + |v_{\mathbf{n}}|^2 = \frac{\epsilon_{\mathbf{n}}}{\omega_{\mathbf{n}}}, \qquad u_{\mathbf{n}}v_{\mathbf{n}}^{*} = -\frac{\zeta_{\mathbf{n}}}{\omega_{\mathbf{n}}}, \tag{A.8}$$

with $\omega_{\mathbf{n}} = 2\mu|p_n \bar{p}_{\bar{n}}|$, we obtain the Hamiltonian in the form of Eq. (54).

# B  Correlation functions

We now want to examine the correlation functions of the model. To start, we notice that the mode expansion for the $\varphi_l$ fields can be written as

$$\varphi_1(\mathbf{r}) = -\sqrt{\frac{\pi}{4L\bar{L}}}\sum_{n,\bar{n}>0}\frac{1}{\sqrt{P_n\bar{P}_{\bar{n}}}}\left\{\left[U_1(\mathbf{n})b_{-n,-\bar{n}}^{\dagger} + U_1^{*}(\mathbf{n})b_{n,\bar{n}}\right]e^{ip_n\xi^z + i\bar{p}_{\bar{n}}\bar{\xi}^z}\right. \tag{B.1}$$

$$\left. + \left[U_1(\mathbf{n})b_{-n,\bar{n}}^{\dagger} + U_1^{*}(\mathbf{n})b_{n,-\bar{n}}\right]e^{ip_n\xi^z - i\bar{p}_{\bar{n}}\bar{\xi}^z} + \mathrm{h.c.}\right\},$$

$$\varphi_2(\mathbf{r}) = -i\sqrt{\frac{\pi}{4L\bar{L}}}\sum_{n,\bar{n}>0}\frac{1}{\sqrt{P_n\bar{P}_{\bar{n}}}}\left\{\left[U_2^{*}(\mathbf{n})b_{n,\bar{n}} - U_2(\mathbf{n})b_{-n,-\bar{n}}^{\dagger}\right]e^{ip_n\xi^z + i\bar{p}_{\bar{n}}\bar{\xi}^z}\right. \tag{B.2}$$

$$\left. - \left[U_2^{*}(\mathbf{n})b_{n,-\bar{n}} - U_2(\mathbf{n})b_{-n,\bar{n}}^{\dagger}\right]e^{ip_n\xi^z - i\bar{p}_{\bar{n}}\bar{\xi}^z} - \mathrm{h.c.}\right\},$$

where the quantities $U_l(\mathbf{n})$ are given in terms of the coefficients of the Bogoliubov transformation:

$$U_l(\mathbf{n}) = u_{\mathbf{n}} - (-1)^l v_{\mathbf{n}}. \tag{B.3}$$

Note that $u_{\mathbf{n}}$ and $v_{\mathbf{n}}$ only depend on the sign of $n\bar{n}$ through Eq. (A.6) since the dependence on $|p_n\bar{p}_{\bar{n}}|$ cancels out in Eq. (A.8).

The mode expansion can be brought into a continuum form with the replacements

$$\frac{1}{L\bar{L}}\sum_{\mathbf{n}} \to \int \frac{d^2p}{(2\pi)^2}, \quad \delta_{\mathbf{nm}} \to \frac{(2\pi)^2}{L\bar{L}}\delta(\mathbf{p}-\mathbf{p}'), \quad b_{\mathbf{n}} \to \frac{1}{\sqrt{L\bar{L}}}b_{\mathbf{p}}. \tag{B.4}$$

This renders the mode expansion as

$$\varphi_1(\mathbf{r}) = -\sqrt{\frac{\pi}{4}}\int_{p,\bar{p}>0}\frac{d^2p}{(2\pi)^2\sqrt{p\bar{p}}}\left\{\left[U_1(\mathbf{p})b^\dagger_{-p,-\bar{p}} + U_1^*(\mathbf{p})b_{p,\bar{p}}\right]e^{ip\xi^z+i\bar{p}\bar{\xi}^z}\right. \tag{B.5}$$

$$\left. +\left[U_1(\mathbf{p})b^\dagger_{-p,\bar{p}} + U_1^*(\mathbf{p})b_{p,-\bar{p}}\right]e^{ip\xi^z-i\bar{p}\bar{\xi}^z} + \text{h.c.}\right\},$$

$$\varphi_2(\mathbf{r}) = -i\sqrt{\frac{\pi}{4}}\int_{p,\bar{p}>0}\frac{d^2p}{(2\pi)^2\sqrt{p\bar{p}}}\left\{\left[U_2^*(\mathbf{p})b_{p,\bar{p}} - U_2(\mathbf{p})b^\dagger_{-p,-\bar{p}}\right]e^{ip\xi^z+i\bar{p}\bar{\xi}^z}\right. \tag{B.6}$$

$$\left. -\left[U_2^*(\mathbf{p})b_{p,-\bar{p}} - U_2(\mathbf{p})b^\dagger_{-p,\bar{p}}\right]e^{ip\xi^z-i\bar{p}\bar{\xi}^z} - \text{h.c.}\right\}.$$

From the above expressions we can compute the associated equal-time correlation functions of the fields $\varphi_l(\mathbf{r})$. All components of the correlation matrix are proportional to the same integral:

$$\langle\varphi_l(\mathbf{r})\varphi_{l'}(0)\rangle = \frac{\mathcal{C}_{ll'}}{16\pi}\int_{p,\bar{p}>0}\frac{d^2p}{p\bar{p}}\left(e^{ip\,\xi^z+i\bar{p}\bar{\xi}^z} + e^{ip\,\xi^z-i\bar{p}\,\bar{\xi}^z} + \text{h.c.}\right)e^{-\alpha_s(p+\bar{p})}, \tag{B.7}$$

where $\mathcal{C}_{ll'}$ are the elements of the Hermitian matrix

$$\mathcal{C} = \begin{pmatrix} \frac{M_{22}}{\mu} & -i + \frac{M_{12}}{\mu} \\ i + \frac{M_{12}}{\mu} & \frac{M_{11}}{\mu} \end{pmatrix}, \tag{B.8}$$

and $\alpha_s$ is a short-distance cutoff. Solving the integrals, we find that the correlations have a double logarithmic behavior

$$\langle\varphi_l(\mathbf{r})\varphi_{l'}(0)\rangle = \frac{\mathcal{C}_{ll'}}{16\pi}\left(\ln\left\{p_c^2\left[\alpha_s^2 + (\xi^z)^2\right]\right\}\ln\left\{p_c^2\left[\alpha_s^2 + (\bar{\xi}^z)^2\right]\right\}\right). \tag{B.9}$$

Here we have introduce a small-momentum cutoff $p_c$ to regularize the infrared divergence of the integral. We must take $p_c \to 0$ at the end of the calculation of correlation functions of local operators. We can also calculate the correlation of the derivative of the fields, which are associated with dipoles. In particular, we have

$$\left\langle\partial_{\xi^z}\varphi_l(\xi^z = 0, \bar{\xi}^z)\partial_{\xi^z}\varphi_{l'}(0,0)\right\rangle = -\frac{\mathcal{C}_{ll'}}{8\pi\alpha_s^2}\ln\left\{p_c^2\left[\alpha_s^2 + (\bar{\xi}^z)^2\right]\right\}. \tag{B.10}$$

We obtain a similar expression for $\langle\partial_{\bar{\xi}^z}\varphi_l(\mathbf{r})\partial_{\bar{\xi}^z}\varphi_{l'}(0)\rangle$ by exchanging $\xi^z \leftrightarrow \bar{\xi}^z$. Quadrupoles are associated with the second derivative $D_{xy}\varphi_l$. In this case, we obtain a $p_c$-independent term in the correlator that decays as a power law at large distances. For $|x^2 - y^2| \gg \alpha_s^2$, we obtain

$$\left\langle D_{xy}\varphi_l(\mathbf{r})D_{xy}\varphi_{l'}(0)\right\rangle \sim \frac{4\mathcal{C}_{ll'}}{\pi(x^2-y^2)^2}. \tag{B.11}$$

Along the directions $y = \pm x$, the correlator changes sign and decays more slowly:

$$\left\langle D_{xy}\varphi_l(x, y = \pm x)D_{xy}\varphi_{l'}(0)\right\rangle \sim -\frac{\mathcal{C}_{ll'}}{2\pi\alpha_s^2 x^2}. \tag{B.12}$$

We are now in position to discuss the correlations of charged operators. To recover the neutrality condition within a direct calculation of correlators of vertex operators, one must be careful with the dependence on the infrared cutoff $p_c$ in the field propagators [62]. We are interested in correlators of the form

$$\left\langle e^{iq_l \varphi_l(\mathbf{r})} e^{-i\tilde{q}_{l'} \varphi_{l'}(0)} \right\rangle = e^{q_l \tilde{q}_{l'} \left\langle \varphi_l(\mathbf{r})\varphi_{l'}(0)\right\rangle - \frac{q_l q_{l'}}{2} \left\langle \varphi_l(\mathbf{r})\varphi_{l'}(\mathbf{r})\right\rangle - \frac{\tilde{q}_l \tilde{q}_{l'}}{2} \left\langle \varphi_l(0)\varphi_{l'}(0)\right\rangle} . \tag{B.13}$$

Using the result in Eq. (B.9), we obtain

$$
\begin{aligned}
\left\langle e^{iq_l \varphi_l(\mathbf{r})} e^{-i\tilde{q}_{l'} \varphi_{l'}(0)} \right\rangle &= \exp\Bigg\{ -\frac{\mathcal{C}_{ll'}}{32\pi}\Bigg[ (q_l - \tilde{q}_l)(q_{l'} - \tilde{q}_{l'}) \ln^2\left(p_c \alpha_s\right) \\
&\quad - 2q_l \tilde{q}_{l'} \ln\left(p_c^2 \alpha_s^2\right) \ln\left( \frac{\alpha_s^4 + \left(\xi^z \bar{\xi}^z\right)^4}{\alpha_s^4} \right) \\
&\quad - 2q_l \tilde{q}_{l'} \ln\left( \frac{\alpha_s^2 + (\xi^z)^2}{\alpha_s^2} \right) \ln\left( \frac{\alpha_s^2 + \left(\bar{\xi}^z\right)^2}{\alpha_s^2} \right) \Bigg] \Bigg\} .
\end{aligned}
\tag{B.14}
$$

We can now see that the double logarithmic behavior is responsible for the vanishing of the correlator for monopoles. Even if we impose the neutrality condition $\tilde{q}_l = q_l$, the argument of the exponential on the right-hand side of Eq. (B.14) retains a dependence on $p_c$ that forces the correlator to vanish when we take the limit $p_c \to 0$. As a result, the monopole operators are always short-range correlated. As discussed in Sec. 4.2, this type of correlation could only be nonzero if the generalized neutrality condition in Eq. (61) were satisfied, which is not the case for two monopoles created at different positions.

On the other hand, the correlator of dipoles does not share this problem. The reason is that $\langle \partial_{\xi^z} \varphi_l(\mathbf{r}) \partial_{\xi^z} \varphi_{l'}(0) \rangle$ scales as a simple logarithm, see Eq. (B.10). Performing the same analysis with regularized propagators, we obtain a power-law decay along the direction in which the dipoles can move:

$$\left\langle e^{iwq_l \partial_{\xi^z} \varphi_l(\xi^z = 0, \bar{\xi}^z)} e^{-iw\tilde{q}_{l'} \partial_{\xi^z} \varphi_l(0)} \right\rangle = (p_c \alpha_s)^{\frac{w^2 \mathcal{C}_{ll'}(q_l - \tilde{q}_l)(q_{l'} - \tilde{q}_{l'})}{8\pi \alpha_s^2}} \left[ \frac{1}{\alpha_s^2 + \left(\bar{\xi}^z\right)^2} \right]^{\frac{w^2 \mathcal{C}_{ll'} q_l \tilde{q}_{l'}}{8\pi \alpha_s^2}} . \tag{B.15}$$

Setting $\tilde{q}_l = q_l$ is a sufficient condition to prevent this correlator from vanishing in the limit $p_c \to 0$. For $\tilde{q} = q \in \mathbb{Z}^2$, the imaginary part in the off-diagonal matrix elements of $\mathcal{C}$ does not contribute to the exponent, and we obtain the result in Eq. (74) of the main text.

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
