# Peer review of "Boundary Modes in the Chamon Model"

_SciPost Physics, doi:SciPost Phys. 15, 010 (2023)_

## Round 1 · Referee Report · Anonymous (Referee 1) · 2023-2-5

Strengths

  1. interesting investigation of boundary phases in fractonic models

  2. impressive field theoretic description

  3. intriguing argument for a possible stable gapless boundary phase in the weak coupling regime

Weaknesses

  1. too limited results at this stage to gauge the importance that the prediction of a stable gapless boundary phase may have in the field

  2. the novelty/importance for the broader readership of the technique deployed is not sufficient to stand on its own as a criterion for publication

Report

This paper studies the boundary properties of the Chamon model. It begins with a characterisation of the behaviour at the microscopic lattice level, and then proceeds to deploy a field theoretic description, which is useful in particular to gain insight on the possible behaviour beyond the strong coupling limit appropriate for the microscopic model. The authors point out the possible existence of a stable gapless boundary phase in the weak coupling regime.

I am in two minds about recommending this paper for publication in SciPost and I would like to express my thoughts below, for the authors' and the editor's consideration.

At the microscopic level, the work is new but the results are in some way expected, and it is not immediately clear to me that one can learn something significantly novel from them (please do correct me if I am wrong -- in which case, it may help to stress is more clearly in the manuscript). At the field theoretic level, the authors' work is definitely an impressive tour de force. It certainly deserves credit and publication. However, as I understand it, the technique that is employed was already introduced and used in earlier work (which the authors correctly and openly credit); the main innovation there is the use of a layer index (which is indeed important and significant). The approach is successful in producing a boundary theory that is consistent with expectations from results in related models (e.g., the X-cube, as the authors themselves point out). The behaviour of the system inferred from field theory is consistent with the microscopic lattice model in the corresponding strong coupling regime. Therefore, the main advantage of and novel result that derives from the field theory is the ability to speculate about the weak coupling regime, where the authors argue that a stable gapless boundary phase is possible.

I am in two minds about the general interest that this manuscript may raise and therefore I hesitate to recommend it for publication in SciPost. It should however certainly be published somewhere, and for reference I personally would consider this a shoe-in for PRB. This is my personal conclusion. I would however be very happy for the editor to make an informed opposite decision, or for the authors to provide further arguments, and perhaps correspondingly improve the discussion/intro sections of the manuscript.

With respect to the SciPost criteria:

1) "Expectations (at least one required) - the paper must:"

At the moment, I would consider that the paper possibly opens a new pathway in the study of boundary behaviour in fractonic models, but the case is marginal and I do not see clear potential for multipronged follow-up work.

At the moment, I do not see reasons for any of the other criteria in this section to be met.

2) "General acceptance criteria (all required) - the paper must:"

All criterial are thoroughly met.

Requested changes

On a separate note, here are a few comments for the authors, in order of appearance in the manuscript:

1) typo: Ref.8 and 40 are the same paper. (I have not checked all other refs.)

2) shortly after Eq.(3), and throughout the paper, the authors make use of a nomenclature derived from gauge theories (namely, they talk about monopoles, dipoles, quadrupoles). This nomenclature is suggestive (and often used in this field of research, and I apologise if the answer to my comment happens to be trivial), but I see no explanation in the manuscript as to why one should use it and why it is useful. For instance, one could use this terminology in the toric code, but it is not helpful and it is not done, to the best of my knowledge (it is a Z_2 gauge theory). The nomenclature is on the contrary used and useful in dimer and vertex models, where the emergent theory is U(1), and indeed electromagnetic-like. Is there a notion of conserved gauge flux and charge-current continuity in this model that supports the use of such notation? If this is the case, it may help to explain it briefly and/or refer to some relevant literature. If there isn't, then perhaps pairs and quadruples of defects are a more appropriate naming than dipoles and quadrupoles.

3) in Fig.2 it may be helpful for the reader if the authors highlighted the sites on which the sigma^x operators are acting to produce the resulting excitation structure.

4) typo: on p.4, there is a missing "if" in the sentence: "... if *** is even or to C and D sub-lattices (if) *** is odd".

5) on p.4, when local cage-net tetrahedron operators are defined and argued to commute with the Hamiltonian, I wonder if it would be helpful to add a brief paragraph discussing the residual degrees of freedom in the ground state of the system, and showing that they are fully resolved (with the addition of appropriate topological operators, I think). This would complete nicely the discussion. At the moment, the reader is left wondering whether we have identified a sufficient number of conserved quantities to resolve the residual 2^(N/2) GS degeneracy after Eq.(3) has been imposed.

6) in addition to the boundary operator in Eq.(6), one could also define in principle another broken stabilizer centred at r=(i,j,1). This would be made up of a single sigma^z_{r-e3} operator, I think. Is there a reason why such additional boundary stabilizers are not considered?

7) incidentally, the choice r-e3 in Eq.(6) suggests that the z<=0 half-space is occupied by the system, and z>0 is empty. I think it would be good to state this choice explicitly in the text (some readers may instinctively put the system in the z>=0 half-space).

8) on p.5 the authors introduce the concept of boundary fracton. Could this concept be explained in a little more detail? Is a boundary fracton different from a bulk one, or can it be simply thought of as a fracton that happens to have reached the boundary?

Also, in relation to my point 2) above, it is not obvious to me that the notion of boundary fracton can be made consistent with a gauge charge interpretation of the defects. I presume it is so; in which case adding a few lines to explain and clarify the point would help, I think.

9) around Eq.(17) it may be worth clarifying to the reader that the dimensionless vector \vec{r} has been replaced with a_s \vec{r}. It is a trivial point, but worth stating, in my opinion.

10) typo: just below Eq.(24): "the the electric".

11) Eq.(26) is argued to follow from Eq.(25) in the absence of matter. Could this be explained in greater detail / referenced if need be? The fact that the notion of matter has not been introduced at all in the paper suggests that a certain level of expertise and familiarity with gauge theories is required to derive (26) from (25).

12) in Eq.(27), I wonder if a condition I \neq J \neq K is needed, of if the expression works in any case.

13) typo, below Eq.(43): "by the ever number of" should be "even", I think

(I must admit that I have not attempted to re-derive all the steps in the equations in Sec.4)

  • validity: high
  • significance: good
  • originality: good
  • clarity: good
  • formatting: excellent
  • grammar: perfect

Author:  Weslei Fontana  on 2023-03-09  [id 3462]

(in reply to Report 1 on 2023-02-05)
Category:
answer to question

We thank the referee for the careful reading of our manuscript and for raising important points which have helped us improve the presentation, emphasizing the novelty of our results. First, we would like to address some general remarks in the referee’s report:

"At the microscopic level, the work is new but the results are in some way expected, and it is not immediately clear to me that one can learn something significantly novel from them (please do correct me if I am wrong – in which case, it may help to stress is more clearly in the manuscript)."

We would like to clarify that we actually did learn something novel by analyzing the microscopic model in section II. Based only on previous work on the Chamon model with periodic boundary conditions, it was not clear whether the model with a (001) surface could have different types of gapped boundaries separated by boundary phase transitions. Here it is instructive to compare with the boundaries of the toric code in 2D (see Bravyi and Kitaev, Ref. [33]) and the X-cube model in 3D (see Bulmash and Iadecola, Ref. [41]), where smooth and rough boundaries are clearly geometrically distinct and are associated with the condensation of different types of particles. Despite some similarities with the X-cube model, the Chamon model exhibits different global constraints which have an impact on its bulk and boundary physics. In fact, it was important to go through the analysis of the microscopic model to appreciate the role of the layer index in distinguishing between the two types of gapped boundaries. Moreover, this analysis taught us that, unlike the X-cube model, there is no condensation of pairs of defects at the (001) boundary of the Chamon model; rather, the boundary allows for the conversion of a pair of defects into a single defect. This understanding developed in the exactly solvable model was crucial for our formulation of the boundary conditions in the effective field theory developed in later sections. Following the referee’s remark, we have rewritten the introduction to emphasize what we learn from the microscopic model in section II and how the results differ from the expectation based on previous results in related models.

"At the field-theoretic level, the authors’ work is definitely an impressive tour de force. It certainly deserves credit and publication. However, as I understand it, the technique that is employed was already introduced and used in earlier work (which the authors correctly and openly credit); the main innovation there is the use of a layer index (which is indeed important and significant). The approach is successful in producing a boundary theory that is consistent with expectations from results in related models (e.g., the X-cube, as the authors themselves point out). The behavior of the system inferred from field theory is consistent with the microscopic lattice model in the corresponding strong coupling regime. Therefore, the main advantage of and novel result that derives from the field theory is the ability to speculate about the weak coupling regime, where the authors argue that a stable gapless boundary phase is possible."

We agree that our prediction of a gapless boundary phase is the main advantage of the effective field theory in comparison with the exactly solvable lattice model. While we have employed field theory techniques introduced in Ref. [20], an important contribution of our work was to examine the boundary processes within this approach, for instance, what happens to the line operators that terminate at the open boundary. Here there are also important differences between our results and previous results on the X-cube model, notably the boundary conditions encoded in Eq. (38).

** “I am in two minds about the general interest that this manuscript may raise and therefore I hesitate to recommend it for publication in SciPost. It should however certainly be published somewhere, and for reference, I personally would consider this a shoe-in for PRB. This is my personal conclusion. I would however be very happy for the editor to make an informed opposite decision, or for the authors to provide further arguments, and perhaps correspondingly improve the discussion/intro sections of the manuscript. With respect to the SciPost criteria: 1) "Expectations (at least one required) - the paper must:" At the moment, I would consider that the paper possibly opens a new pathway in the study of boundary behavior in fractonic models, but the case is marginal and I do not see clear potential for multipronged follow-up work.” **

We believe our work meets the SciPost criteria because it has the potential to open a new research direction and spark a follow-up investigation. To the best of our knowledge, we were the first to speculate about the possibility of a gapless boundary phase in fractonic systems using the effective theory, despite the challenges posed by UV/IR mixing in these models. To further explore this result, it is crucial to build a corresponding phase diagram using microscopic models. Studies of quantum phase transitions in the bulk of fractonic systems are relatively scarce and recent; see for instance Pretko and Radzihovsky, PRL 121, 235301 (2018), and Muhlhauser et al., PRB 101, 054426 (2020). A natural follow-up to our work is to study critical phenomena at the boundary of fractonic systems, without destroying the fracton topological order in the bulk, and to investigate whether and how the associated boundary phase transitions differ from quantum phase transitions in two-dimensional systems. In fact, we are currently working on this problem with the goal of approaching the gapless boundary phase by adding suitable boundary perturbations to the lattice model. While so far we have focused our efforts on the (001) boundary of the Chamon model, this problem is more general and can be very rich since different fracton models have different particle contents and the boundary phases may also depend on the geometry of the boundary. We have expanded the manuscript, in particular the introduction and conclusion sections, to further illustrate our results and the avenues it opens up for future investigations.

Let us now address the list of requested changes. We thank the referee for spotting the typos listed in points 1, 4, 10, and 13 of the report. We have corrected them in the revised version of our manuscript. Concerning the other points:

** “shortly after Eq.(3), and throughout the paper, the authors make use of a nomenclature derived from gauge theories (namely, they talk about monopoles, dipoles, and quadrupoles). This nomenclature is suggestive (and often used in this field of re-search, and I apologize if the answer to my comment happens to be trivial), but I see no explanation in the manuscript as to why one should use it and why it is useful. For instance, one could use this terminology in the toric code, but it is not helpful and it is not done, to the best of my knowledge (it is a Z2 gauge theory). The nomenclature is on the contrary used and useful in dimer and vertex models, where the emergent theory is U(1), and indeed electromagnetic-like. Is there a notion of conserved gauge flux and charge-current continuity in this model that supports the use of such notation? If this is the case, it may help to explain it briefly and/or refer to some relevant literature. If there isn’t, then perhaps pairs and quadruples of defects are a more appropriate naming than dipoles and quadrupoles.”**

Indeed we adopted the nomenclature used in the literature without much explanation. The excitations of the Chamon model were referred to as “monopoles”, “dipoles” and “quadrupoles” in Ref. [2] even before the term “fracton” was coined in Ref. [3]. As the referee correctly points out, this nomenclature is better motivated for effective field theories with emergent U(1) symmetries. This is actually the case of effective theories for fracton models, which contain conservation laws described by continuity equations with higher derivatives. In our case, if we take the action in Eq. (25) and couple the gauge fields to matter (representing excitations above the ground state manifold) respecting gauge invariance, we obtain a continuity equation of the form

$$ \partial_0\,\rho^{(l)}=D_{xz} j^{(l)}+D_{yz} j^{(l)},~~~~ $$
with $\rho^{(l)}$ and $j^{(l)}$ the corresponding charge density and currents. This equation implies conservation of charge as well as dipole moments such as $P_a^{(l)} = \int d^3r \xi^a \rho^{(l)}$. The conservation of the dipole moment is related to the mobility of pairs of defects discussed for the lattice model in Ref. [2]. The constraints of the model imply that two defects in neighboring octahedra (or dipole) cannot annihilate each other and such a pair can only move along the direction of a rigid string. On the hand, quadruples of defects (quadrupoles for short) can be created from the vacuum and have no mobility constraints. Thus, the nomenclature is helpful in the sense that it conveys some information about the mobility and decay of the composite excitations. By contrast, in the toric code, there are no mobility constraints, pairs of defects on neighboring plaquettes belong to the vacuum superselection sector, and this nomenclature is not useful. We agree with the referee that this point deserves clarification. In the revised version of our manuscript we explicitly mention that we follow the nomenclature of Ref. [2] and that the latter is further motivated by the emergent conservation laws within the effective field theory. We also added references to the reviews by Nandkishore and Hermele [Annu. Rev. Condens. Matter Phys. 10, 295 (2019)] and Pretko et al. [Int. J. Mod. Phys. A 35(06), 2030003 (2020)], where the role of dipoles in lattice models and effective field theories for fractons is discussed in more detail.

“in Fig.2 it may be helpful for the reader if the authors highlighted the sites on which the σx operators are acting to produce the resulting excitation structure.”

Following the referee’s suggestion, we have modified Fig. 2 to highlight the sites where we apply the spin operators to create the defects.

** “on p.4, when local cage-net tetrahedron operators are defined and argued to commute with the Hamiltonian, I wonder if it would be helpful to add a brief paragraph discussing the residual degrees of freedom in the ground state of the system, and showing that they are fully resolved (with the addition of appropriate topological operators, I think). This would complete nicely the discussion. At the moment, the reader is left wondering whether we have identified a sufficient number of conserved quantities to resolve the residual 2N/2 GS degeneracy after Eq.(3) has been imposed.”**

The cage-net operators indeed commute with the Hamiltonian, but not necessarily among themselves. Thus, finding the ground state degeneracy through the cage-net operators is not so obvious. In Ref. [2] the ground state degeneracy of the Chamon model in the 3-torus geometry was determined by counting the number of independent line operators of the form

$$ W^{(k)}{IJ} = \prod $$}\in \gamma}\sigma_{\mathbf{r}}^{(k)
where $\gamma$ is a rigid closed line, in the $IJ$-plane (with $I\neq J\neq k$), that winds around the system. These operators commute with the Hamiltonian and obey the constraint $W^x_{yz} W^y_{xz} W^z_{xy} = 1$. Following the referee’s suggestion, we have added a discussion regarding the ground state degeneracy and the line operators below Eq. (3).

** “in addition to the boundary operator in Eq.(6), one could also define in principle another broken stabilizer centered at $r=(i,j,1)$. This would be made up of a single $\sigma_{r-e_3}^z$ operator, I think. Is there a reason why such additional boundary stabilizers are not considered?” **

We cannot add the single-spin operator suggested by the referee because each $\sigma^z_{r−e_3}$ anti commutes with four stabilizers on the same plane, meaning that the resulting model would no longer be a stabilizer code. Moreover, the model that we consider already contains one stabilizer for each spin on the boundary. A more physical way to understand the boundary stabilizers is to imagine that we start from the model in the thermodynamic limit and apply a strong magnetic field in the z direction in the half-space $z > 0$. The effective Hamiltonian is then obtained by projecting the stabilizers in the z ≤ 0 region onto the sector where all spins in the z > 0 region are polarized with $\sigma^z_r = +1$. On the boundary plane, this projection reduces the standard six-site stabilizers in Eq. (1) to the five-site stabilizers in Eq. (6). This way of obtaining the Hamiltonian in half space in terms of a projection of the stabilizers was mentioned in section 4.4. Motivated by the referee’s question, we have decided to present this argument right after Eq. (6) to clarify our definition of boundary stabilizers.

“incidentally, the choice $r-e_3$ in Eq.(6) suggests that the z<=0 half-space is occupied by the system, and $z>0$ is empty. I think it would be good to state this choice explicitly in the text (some readers may instinctively put the system in the z>=0 half-space).”

That was indeed our choice. We now state it explicitly above Eq. (6).

** “on p.5 the authors introduce the concept of boundary fracton. Could this concept be explained in a little more detail? Is a boundary fracton different from a bulk one, or can it be simply thought of as a fracton that happens to have reached the boundary? Also, in relation to my point 2) above, it is not obvious to me that the notion of boundary fracton can be made consistent with a gauge charge interpretation of the defects. I presume it is so; in which case adding a few lines to explain and clarify the point would help, I think.” **

By boundary fracton we mean a defect of the five-site stabilizers defined on the boundary. Like a bulk fracton, a boundary fracton cannot move without creating additional excitations. However, there are processes involving the boundary fracton that cannot happen in the bulk, for instance, the conversion of a dipole into a boundary fracton illustrated in Fig. 5. This type of process can be consistently incorporated into the effective field theory in terms of the boundary condition in Eq. (38), which connects the bosonic field with its derivative in the direction perpendicular to the boundary. In section 4.3 we used this boundary condition to show that the line operators that have dipoles at their endpoints in the bulk (see our response to point 2 above) actually create a fracton when they reach the boundary. We have added the precise definition of boundary fracton below Eq. (6).

“around Eq.(17) it may be worth clarifying to the reader that the dimensionless vector $\vec{r}$ has been replaced with $a_s\vec{r}$. It is a trivial point, but worth stating, in my opinion.”

We thank the referee for noticing this point. We have decided to include the factor of as in the original definition of the lattice vectors above Eq. (1) to avoid the change in notation.

“Eq.(26) is argued to follow from Eq.(25) in the absence of matter. Could this be explained in greater detail / referenced if need be? The fact that the notion of matter has not been introduced at all in the paper suggests that a certain level of expertise and familiarity with gauge theories is required to derive (26) from (25).”

Here the absence of matter means that we are dealing with the free equations of motion in the gauge theory that describes the ground state subspace. More generally, we could couple the gauge fields to matter currents; see our response to point 2) above. We have changed the sentence to “From the equation of motion...” in order to avoid any confusion.

“in Eq.(27), I wonder if a condition $I\neq J\neq K$ is needed, or if the expression works in any case.”

The condition $I\neq J\neq K$ is indeed needed. We now state this explicitly in Eq. (27).

---

## Round 1 · Referee Report · Andreas Karch (Referee 2) · 2023-2-13

Strengths

1- Apply QFT to fracton model to learn about new phenomena 2- Novel strategy to deal with boundary actions that contain normal derivatives, potentially paving way for many future applications

Weaknesses

1- It's about fractons. So far this is fantasy land, not the real world

Report

This work follows a very similar strategy as myself and my collaborators have used in previous work, their ref [43]. I believe this is a valid and interesting line of research, but obviously I am biased.

Fractons are solvable lattice models. While so far experimental realizations are still lacking, they are very interesting from the theoretical point of view. One reason for this is that they force us to rethink what we mean by a quantum field theory (QFT).

Novel QFTs have been developed to deal with fractons, reproducing many results that hat previously been obtained on the lattice. Studying systems with boundaries is one way to extend these studies. In this case it appears that often the field theory is simpler to analyze than the lattice. Also, studying boundaries proved very fruitful in the case of topological phases of matter. The verdict still seems to be out as to how much we can learn from boundaries in the case of fractons.

The two main novelties of the current work seem to be that 1- they apply it to a different fracton system and 2- at the technical level, they propose a new way of dealing with normal derivatives appearing in the boundary action. They propose that they have to be expressed as linear combinations of boundary fields. This sounds very simple yet correct and may be the key insight to describe other fracton boundaries, for example boundaries of the X-cube along planes other than the trivial (0,0,1) plane we considered in our work.

Overall I think this is a valuable addition to the literature and meets the criteria you require for publication.

Requested changes

1- optional: I would like to see a discussion of how their boundary action is related to ours. Clearly, in the bulk, Chamon's model and X-cube are very different. Yet the boundary theories looks similar. What are the differences?

  • validity: high
  • significance: good
  • originality: high
  • clarity: good
  • formatting: excellent
  • grammar: excellent

Author:  Weslei Fontana  on 2023-03-09  [id 3461]

(in reply to Report 2 by Andreas Karch on 2023-02-13)
Category:
answer to question

We thank Prof. Karch for the careful reading of our manuscript and for stating that our work meets the SciPost criteria for publication.
We agree that our proposal to handle the normal derivatives in the boundary conditions should be useful to analyze other boundaries of related fracton models. In fact, it is interesting to note that the (110) boundary of the X-cube model discussed by Bulmash and Iadecola in Ref. [41] shares some properties with the (001) boundary Chamon model.
Concerning the question in the “requested changes”, indeed the boundary theory for the X-cube model derived in Ref. [43] is similar to the one for the Chamon model since both are written in terms of a two-component bosonic field with a $2 \times 2$ antisymmetric K matrix. Here it seems important to note that the BF theory for the X-cube model contains more gauge fields, but the constraints and the $S_4$ symmetry allow one to express the boundary action in terms of only two components. In the Chamon model, the two fields are associated with the layer index from the beginning. In any case, our current understanding is that different fracton models can have the same boundary theory because the anomaly of the boundary does not uniquely determine the bulk. This statement was verified for the X-cube model in Ref. [43] and it is consistent with the anomaly inflow in the presence of subsystem symmetries discussed by Burnell et al. in Phys. Rev. B 106, 085113 (2022).

---

## Round 1 · Referee Report · Anonymous (Referee 1) · 2023-3-25

Report

I thank the referes for their detailed responses to my comments and criticisms. Upon further consideration, I am happy to change my view and to recommend this manuscript for publication.

---

## Round 1 · Referee Report · Anonymous (Referee 3) · 2023-4-10

Report

This paper contains a study of the boundary of the Chamon fracton model, within a particular field theory approach. The study appears to be done carefully, with many details given. The continuum boundary theory is able to recover the gapped boundaries of the lattice model, and the authors also suggest it can describe a stable gapless boundary phase.

As a general matter, there are many not-yet-understood conceptual subtleties involved in using continuum theories as effective theories for fracton lattice models. So I cannot be entirely confident that the authors' continuum theory correctly describes possible boundaries of the lattice model. However, I think sorting this out to my satisfaction would be too much to ask of the authors of this paper (not to mention too much to ask of myself as a referee) -- figuring this out is a research project in its own right. Nonetheless, I think this paper is a valuable contribution to the literature on fracton models and continuum theories thereof, and given the careful analysis presented it will provide useful information as the community continues to work on sorting out some of the challenging underlying issues. Therefore I am happy to recommend this paper for publication in SciPost Physics.

---

## Round 2 · List of Changes

• Corrected the typos pointed out by Referee 1.
• Modified Fig. 2 to highlight the sites where the spin operators are applied.
• Updated Refs. [30,32,43,53,56,59], which are now published.
• Added citations to J. Sous and M. Pretko, npj Quantum Materials (2020); J. Sous and M. Pretko, Phys. Rev. B 102, 214437 (2020); W. Shirley, X. Liu and A. Dua, Phys. Rev. B 107(3) (2023); R. M. Nandkishore and M. Hermele, Annu. Rev. Condens. Matt. Phys. 10, 295 (2019); M. Pretko, X. Chen and Y. You, Int. J. Mod. Phys. A 35, 2030003 (2020); S. Liu and W. Ji, Towards Non-Invertible Anomalies from Generalized Ising Models, arXiv: 2208.09101 (2022).
• Expanded to introduction to elaborate on the comparison with related models and emphasize our main results.
• Included the factor of as in the lattice vectors above Eq. (1).
• Added a discussion about the ground state degeneracy below Eq. (3).
• Added an explanation about the nomenclature of multipoles on page 5.
• Included the definition of boundary fracton and further discussion about the boundary stabilizers below Eq. (6).
• Corrected factors of 2 in Eqs. (57), (58), and (71); these factors do not affect our conclusions.
• Rewrote Eq. (76) to give a more general expression for the exponents for cosine operators
with non-elementary charges.
• On page 18, expanded the discussion about the gapless boundary phase.
• On page 19, added remarks about the comparison with the X-cube model.
• Modified Fig. 2 to highlight the sites where the spin operators are applied.
• Updated Refs. [30,32,43,53,56,59], which are now published.
• Added citations to J. Sous and M. Pretko, npj Quantum Materials (2020); J. Sous and M. Pretko, Phys. Rev. B 102, 214437 (2020); W. Shirley, X. Liu and A. Dua, Phys. Rev. B 107(3) (2023); R. M. Nandkishore and M. Hermele, Annu. Rev. Condens. Matt. Phys. 10, 295 (2019); M. Pretko, X. Chen and Y. You, Int. J. Mod. Phys. A 35, 2030003 (2020); S. Liu and W. Ji, Towards Non-Invertible Anomalies from Generalized Ising Models, arXiv: 2208.09101 (2022).
• Expanded to introduction to elaborate on the comparison with related models and emphasize our main results.
• Included the factor of as in the lattice vectors above Eq. (1).
• Added a discussion about the ground state degeneracy below Eq. (3).
• Added an explanation about the nomenclature of multipoles on page 5.
• Included the definition of boundary fracton and further discussion about the boundary stabilizers below Eq. (6).
• Corrected factors of 2 in Eqs. (57), (58), and (71); these factors do not affect our conclusions.
• Rewrote Eq. (76) to give a more general expression for the exponents for cosine operators
with non-elementary charges.
• On page 18, expanded the discussion about the gapless boundary phase.
• On page 19, added remarks about the comparison with the X-cube model.

---

## Round 3 · Author Response

We would like to address the late report we received in the previous resubmission.

Referee "I am troubled by section 3, which is reproducing a field theory that was written down quite some time ago in Ref. 9 (which is also Ref. 27), without proper acknowledgment thereof. (they appear to obtain the same field theory in a slightly different coordinate system, from a different derivation, and Ref. 9 does explicitly make the connection to the Chamon code in Section VII, which is essentially the reverse of the process that they use to derive it). — Similarly, the analysis of the edge in the continuum appears to be quite similar to the discussion in IIIB of Ref. 9. Again, in my view, this section does not properly reference the previous work that did essentially the same thing."

We start by adressing the comment regarding similarities with the work “Fractonic Chern- Simons and BF theories”. There, the authors considered Chern-Simons and BF theories of higher-rank gauge fields and what consequences the exotic gauge structure arising from the higher-order derivatives would bring. They adopted a bottom-up approach (i.e., going from the IR to UV) to make the connection to the Chamon model in a cubic lattice instead of the fcc lattice where the model was originally defined. Before addressing the main differences between the approaches, we would like to apologize for not emphasizing their work in section 3. Comments regarding their work were added to properly acknowledge what has been done and to make more explicit the important differences that we treat in our work. Now to the differences: As mentioned previously, they adopted a bottom-up approach to obtain the Chamon model in the cubic lattice. Although not necessarily problematic, this approach hides some subtleties that are evident in the top-down (UV to IR) approach that we took, for example, highlighting the role that the lattice spacing plays in the field theory. Another crucial and important difference is that the connection to the Chamon model in their work is through a higher rank Chern-Simons theory (meaning that there is only one species of gauge field) that is then discretized into the lattice to obtain the Chamon model in cubic geometry. This again is a consequence of the bottom-up approach. In this case, the layer index that we have considered, which is explicit in the lattice theory, suggests to us that the field theory description should contain two species of gauge fields, hence connecting the Chamon model with a double higher rank Chern-Simons theory (or higher rank BF theory) instead of the single Chern-Simons term obtained previously. Regarding the boundary description, we are not sure about the similarities that the referee pointed out. Given the description in terms of Chern-Simons or BF theories, it is expected that there will be non-trivial modes at the boundary due to the gravitational anomaly that appears when one puts the model into a finite manifold. In their description of the boundary, despite the level (s parameter) being dimensionful and the theory being dynamically trivial, the boundary treated in their work is not equivalent to ours. In our geometry, their case corresponds to a (011) boundary, instead of the (001) boundary we considered. In the tilted geometry that they work, obtained from requiring C3 invariance along the (111) direction, will inevitably lead to a boundary theory with mixed derivatives, which makes their construction more similar to the case of the X-cube model, explored in more detail by Prof. Karch and collaborators (also referenced in our work). In our case, the analysis should be taken with more care since the boundary we have considered contains normal derivatives of the fields that would lead to non-physical extra degrees of freedom at the boundary. Handling these normal derivative terms is an important point of our discussion in order to obtain a consistent theory at the boundary. Although subtle, this is a striking difference between our case and the work the referee mentioned. With all that considered, we believe that all these subtleties are important and constitute the main differences between what we did and what was done previously.

Referee

"The question of why it is that the edges are gappable is indeed a puzzling one when approached from this continuum theory. It’s interesting that the authors write down such gapping terms, but I do not find their analysis very satisfying. In particular, the thing that I found surprising about this gappability is that coming from 2 dimensions, it’s natural to expect that a Chern-Simons-like theory has some kind of gravitational anomaly, such that even if you are willing to break charge conservation you would still have a gapless boundary. I don’t see a discussion of this in their approach; they write down cosine terms without asking whether these should be compatible with the commutation relations of the fields. It may well be that it is and the analysis is correct, but I would certainly have appreciated a more careful justification of this."

We thank the referee for pointing this out. Perhaps the compatibility of the gapping terms was not that explicit in our presentation. Note that even in two dimensions there is the possibility of gapped boundaries. The gravitational anomaly measures the thermal Hall conductance $\kappa_H$, then of course, when one has $\kappa_H\neq 0$ then the boundary is necessarily gapless, but for $\kappa_H =0$ the boundary can be gapped given that the gapping terms obey a list of criteria (see for example Phys. Rev. X 3, 021009 (2013), Phys. Rev. B 91, 125124 (2015)). In any case, our analysis of the gappability of the boundary is in similar reasoning of those in these works we have mentioned. To start, the number of compatible cosines that can be added into the action depends on the size of the K-matrix, for an N × N matrix only N/2 cosines can be added such that gap remains stable. Adding less than that would not gap all the fields, adding more than that would lead to incompatible interactions and hence an unstable spectrum, as we have mentioned in the paragraph near Eq.(71) of the revised manuscript (Previously it was near eq.(74)). This is why we only considered one gapping term, given that our K matrix is 2×2. The “null and mutual statistics” condition (borrowing the terminology of Phys. Rev. B 91, 125124 (2015)) are encoded in our Eq.(71) of this new version, which guarantees that all possible gapping terms will commute, and hence can be pinned altogether. These conditions also ensure the stability of the gapped spectrum.

---

## Round 3 · List of Changes

• Extended discussions in sections 3 and 4.4

---

## Editorial Decision

published